# Inhalation Dosage Forms: A Focus on Dry Powder Inhalers and Their Advancements

**DOI:** 10.3390/ph16121658

**Published:** 2023-11-28

**Authors:** Sabrina Magramane, Kristina Vlahović, Péter Gordon, Nikolett Kállai-Szabó, Romána Zelkó, István Antal, Dóra Farkas

**Affiliations:** 1Department of Pharmaceutics, Semmelweis University, Hőgyes Str. 7, H-1092 Budapest, Hungary; magramane.sabrina@pharma.semmelweis-univ.hu (S.M.); vlahovic.kristina@phd.semmelweis.hu (K.V.); antal.istvan@semmelweis.hu (I.A.); 2Department of Electronics Technology, Budapest University of Technology and Economics, Egry J. Str. 18, H-1111 Budapest, Hungary; gordon.peter@vik.bme.hu; 3Department of Pharmacy Administration, Semmelweis University, Hőgyes Str. 7–9, H-1092 Budapest, Hungary; zelko.romana@semmelweis.hu

**Keywords:** dry powder inhalers, pulmonary delivery, inhalation dosage forms, particle engineering

## Abstract

In this review, an extensive analysis of dry powder inhalers (DPIs) is offered, focusing on their characteristics, formulation, stability, and manufacturing. The advantages of pulmonary delivery were investigated, as well as the significance of the particle size in drug deposition. The preparation of DPI formulations was also comprehensively explored, including physico-chemical characterization of powders, powder processing techniques, and formulation considerations. In addition to manufacturing procedures, testing methods were also discussed, providing insights into the development and evaluation of DPI formulations. This review also explores the design basics and critical attributes specific to DPIs, highlighting the significance of their optimization to achieve an effective inhalation therapy. Additionally, the morphology and stability of 3 DPI capsules (Spiriva, Braltus, and Onbrez) were investigated, offering valuable insights into the properties of these formulations. Altogether, these findings contribute to a deeper understanding of DPIs and their development, performance, and optimization of inhalation dosage forms.

## 1. Introduction

Archives documented the use of inhalation therapy as a method of drug delivery back centuries, with historical records documenting the inhalation of various medicinal vapors and fumes for the treatment of respiratory diseases [1,2]. But it was not until recent decades that inhalation treatment made significant advancements, becoming a popular strategy for treating respiratory conditions like asthma and chronic obstructive pulmonary disease (COPD). Modern inhalation devices have contributed to this trend, with dry powder inhalers (DPIs) dominating the market because of their distinct benefits and potent drug delivery capabilities [3,4]. Inhalation therapy offers unique benefits in the treatment of respiratory disorders, primarily due to its targeted delivery approach. Inhalation ensures rapid drug absorption and deposition at the site of action by directly delivering medications to the lungs [2,5], therefore minimizing systemic exposure and potential systemic side effects, and thus optimizing and improving therapeutic outcomes, patient compliance, and overall quality of life. Among the diverse range of inhalation devices available, DPIs have become an essential choice, attracting considerable interest. The popularity of these devices is due to a number of features that solve the drawbacks and limitations of conventional inhalers, namely nebulizers and metered-dose inhalers (MDIs). DPIs are renowned for their propellant-free formulation, eliminating concerns over environmental impact and propellant-related adverse effects [4,6,7]. Instead, DPIs employ dry powder formulations, which offer an increased chemical stability compared to their liquid-based counterparts [8]. Furthermore, DPIs offer ease of use and require minimal patient coordination during administration, rendering them accessible and suitable for patients of all ages, including children and the elderly. In addition to being breath-actuated, DPIs also do not require the use of a spacer [7,9]. Clear indications of the growing interest in DPI research were obtained from a comprehensive search of the clinicaltrials.gov database, which revealed a notable rise in clinical studies centered around DPIs. Among these clinical trials, 4 are awaiting participant recruitment, while 15 are actively recruiting. Additionally, 1 study is listed as active but not recruiting, 395 have successfully concluded, and 15 were terminated [10].

The success of DPIs, however, hinges on certain aspects, including the physicochemical properties of the drug formulation (moisture sensitiveness), the design and functionality of the inhaler device, and the patient’s inhalation technique [9,11]. Achieving consistent drug delivery in the lungs remains a challenge in DPI development, necessitating ongoing research and innovation to optimize aerosol generation and particle dispersion upon inhalation. Advances in device design and formulation development have played a pivotal role in overcoming these challenges, elevating DPIs to new heights of efficiency and therapeutic performance.

This review aims to offer a comprehensive examination of inhalation therapy, with a primary focus on DPIs and their associated formulations. The scope of this review encompasses the various categories of DPIs available, including single-dose and multi-dose reservoir DPIs, each with distinct powder dispersion mechanisms. Moreover, we will explore the fundamental attributes of DPI design, considering factors such as user-friendliness, dose accuracy, portability, and hygienic considerations. Furthermore, this review will explore the crucial role of formulation technology in DPI development, addressing strategies to optimize drug stability, enhance fine particle fraction (FPF), and achieve consistent dose uniformity. Additionally, we will explore the increasing adoption of DPIs for delivering biotech drugs, as well as the incorporation of lipid-based or polymer-based carriers to enhance drug bioavailability and therapeutic efficacy. By examining the versatile nature of DPI formulation and delivery devices, this review aims to provide valuable insights to the ongoing efforts in optimizing inhalation therapy and ultimately enhancing patient outcomes.

## 2. Pulmonary Delivery: An Overview

### 2.1. Advantages of the Pulmonary Drug Delivery

The formulation of dosage forms intended for pulmonary delivery plays a crucial role in the performance and effectiveness of inhalers and is consequently highly elaborated. The development of inhalation devices and their formulation throughout the years made it possible for the pulmonary route to deliver drugs locally as well as systemically [12,13,14,15]. The drug delivery can be achieved via oral or nasal route, though the former provides a greater drug deposition due to the physiology of the respiratory system.

The pulmonary route of administration finds its significance in the advantageous properties of the lungs, namely their large surface area but also their high permeability. Other relevant benefits of the pulmonary delivery are its non-invasive nature, the absence of first pass metabolism, a low metabolic activity, and a controlled environment regarding the systemic absorption [5]. Moreover, the pulmonary route allows the delivery of a somewhat small amount of active ingredient directly through the respiratory system, which reaches a high local concentration in the airways while simultaneously lessening systemic adverse effects, making the therapeutic ratio greater than that of oral and parenteral routes [2]. The lungs also possess considerably efficient clearance mechanisms which keep undesired environmental particles from entering the human body. However, these clearance processes may also work against the therapeutic effectiveness of an inhaled drug. Another important criterion to consider assessing the efficiency of an inhaled dosage form is the drug deposition in the respiratory system.

### 2.2. Particle Size and Drug Deposition in the Lungs

Drug deposition in the lungs can affect the therapeutic efficiency of a treatment by inhalation and should therefore be taken into consideration. After the successful inhalation of the particles, drug deposition will be the first step and will happen through different processes, mainly inertial impaction, sedimentation, and diffusion (Table 1), but also other mechanisms such as direct interception and electrostatic deposition.

Figure 1 shows these 3 main processes and the associated particle deposition according to the particle size [16,17,18].

Inertial impaction is the main deposition mechanism in the oropharynx and conducting airways as a result of increased air velocity with turbulent flow. Particles considered large with a diameter higher than 5 µm experience inertia and resist the sudden changes in direction and speed, pushing them out of their initial airstream trajectory and, and ultimately impacting the wall of the upper airways. The probability ($P_{I}$) of a particle deviating from the airstream can be calculated (Equation (1)) by [16,17,18,19]:(1)$$P_{I}=1-\frac{2}{\pi }\cos^{-1}\left(\theta \;\cdot \;\;Stk\right)+\frac{1}{\pi }\sin\;[2\;\cos^{-1}\left(\theta \;\cdot \;\;Stk\right)]\;\;\mathrm{for}\;\left(\theta \;\cdot \;Stk\right)<1\;\mathrm{for}\;\left(\theta \;\cdot \;Stk\right)\geq 1,\;P=1$$
where:$P_{I}$ = probability of impactionθ = bending angle (change in the direction of the flow)*Stk* = Stokes number, defined as (Equation (2)):
(2)$$Stk=\frac{{\rho d}^{2}\upsilon }{18µD},$$
where:ρ = particle densityd = particle diameterυ = particle velocityµ = viscosity of fluidD = airway diameter

The amount of drug lost due to this mechanism is one of the main hindrances in the lung deposition while using a dry powder inhaler [20].

Sedimentation is the mechanism which affects particles between 0.5 and 5 µm as they are transported deeper into the lungs, mainly in the bronchi, bronchioles, and alveolar region. The velocity decreases, and so does the likelihood of deposition by impaction. Thus, sedimentation affects particles which exceeded impaction and reached the last 5–6 generations of the lung. The particles begin to settle due to the force of gravity resulting in sedimentation, which intensifies with an increase in particle diameter, mass, residence time and a decrease in flow rate. Particles with a diameter of 3–5 µm reach the tracheobronchial region by sedimentation, while sedimentation as well as diffusion is expected in case of 0.5–3 µm which reach the alveolar region. The probability of particle deposition by sedimentation ($P_{S}$) is expressed by (Equation (3)) [16,17,18]:(3)$$P_{S}=1-{\;e}^{\left(\;\frac{4{\;g\;C\;p\;d}^{2}L\cos ø}{9\;\pi \;µ\;R\;\upsilon }\;\right)},$$
where:$P_{S}$ = probability of sedimentationg = gravitational forceC = Cunningham slip angle correction factorρ = particle densityd = particle diameterL = length of the tubeø = inclination angle relative to gravityµ = viscosity of fluidR = radius of the airwaysυ = particle velocity

Diffusion dominates the deposition process in the lower airways and alveolar region for particles smaller than 0.5 µm due to the Brownian motion: the random movement of particles induced by their collisions with gas particles. Even so, due to their considerably small size, most of the particles are exhaled and merely few of them deposit. The probability of deposition by diffusion ($P_{D}$) is expressed by (Equation (4)) [16,17,18]:(4)$$P_{D}=\sqrt{\frac{2\mathrm{KTC}}{\frac{3\pi \eta d}{R}}},$$
where:$P_{D}$ = probability of diffusionK = Boltzmann’s constantT = absolute temperatureC = Cunningham slip angle correction factorη = viscosity of gasd = particle diameterR = airway diameter

Direct interception is a mechanism which takes place in case of particles with an elongated shape (such as fibers) which causes them to meet the surface of an airway wall [16,17,21].

Electrostatic precipitation is generally considered less significant than the other deposition mechanisms. When close to the airway walls, electrically charged particles are attracted to become neutralized through binding. As a result, the deposition of charged particles may be larger than that of neutral particles.

Overall, the principal mechanisms affecting relatively large particles are inertial impaction and sedimentation, though gravitational sedimentation may become dominant in the small conducting airways. On the other hand, smaller particles (below 0.5 µm) mainly undergo diffusion. When the mass median aerodynamic diameter (MMAD) of inhaled particles is between 5 and 10 µm, the particles deposit in the oropharyngeal region and large conducting airways. When the diameter is between 1 and 5 µm, the deposition occurs in the small airways and alveoli. Of those deposited particles, more than 50% of particles with a diameter of 3 µm deposit in the alveolar region. The fate of particles with a diameter lower than 3 µm is to deposit either in the lower airways (about 80% chance) or in the alveoli (50–60% chance). Therefore, the ideal particle size would be approximately 3 µm [16,22,23,24,25]. In the case of particles which are considered very small (<1 µm), roughly half will deposit in the alveoli whereas the rest will be exhaled [26].

The deposition of the particles is a determinant factor of the efficiency of an inhaled dosage form. A drug dose high enough to reach therapeutic effectiveness should be deposited past the oropharyngeal region in order for the administration to be effective. The deposition can be influenced by processes such as the particle size of the drug administered, the particle properties, but also clearance mechanisms present in the respiratory system [27].

## 3. Particle Engineering Techniques for DPI Formulations

### 3.1. Manufacturing Procedures for DPI Formulations

In order to achieve the desired particle size range in formulations, various manufacturing techniques are used. Micronization, a common top-down process, reduces solid material particles’ diameter to the micrometer or nanometer range, enhancing dissolution rate and bioavailability while optimizing particle size distribution. Jet milling has customarily been used for micronization but may result in particles with an elevated surface energy and strong electrostatic charge, potentially distorting crystalline materials and increasing formulation adhesiveness. In case of low dose active pharmaceutical ingredients (APIs), excipients enhance bulkiness, flow properties, and dose reproducibility. However, in case of high-dose APIs, excipients cannot be added in large amounts since it will eventually increase the number of inhalations required to be taken by the patients to complete a dose, due to an increase in the powder quantity (antibiotics for example). To overcome that, manufacturing methods such as jet milling or spray drying are used to reduce the particles to a respirable size, though they can influence the physico-chemical characteristics of the drug as well as its solid-state properties and physical stability. They engineer amorphous particles which, since water acts as plasticizer, are prone to recrystallization at high relative humidity. Consequently, it is crucial to understand the impact of storage conditions on the physical stability and solid-state properties of the particles [28,29].

#### 3.1.1. Milling

The process of milling is the micronization or breaking of coarse particles down to a smaller respirable size through mechanical energy or shear. Jet milling is a customary technique, notably in the DPIs production, involving different settings such as fluid impact mills, opposed jet mills, spiral jet mills, oval chamber jet mills and fluidized bed opposed jet mills [29]. In jet milling, the micronization of the particles is completed through particle/particle and/or particle/chamber walls impaction using pressurized gas produced by high particle velocity and high energy. Multiple impactions take place, provoking particles fractures and therefore a reduction in their size. This technique is beneficial notably due to its low cost and availability. However, as stated before, the powdered particles produced tend to be extremely cohesive and inclined to agglomerate. One way to surmount these challenges is conditioning, which consist of letting the particles settle after jet milling in an environment with controlled relative humidity and temperature to facilitate the transition from amorphous to crystalline state. The surface coating of the particles with anti-sticking agents after jet milling is also a possibility [29].

Co-milling to improve the stability or effectiveness of the formulation is also an alternative, using additives. In co-milling, at least two compounds are co-processed by the milling equipment to reduce their particle size but also to blend them together homogenously [29]. For instance, when co-milling beclomethasone dipropionate (BDP) with lactose, the addition of magnesium stearate decreased the adhesiveness between the two components, thereby reducing agglomeration [29,30].

Wet milling is another particle size reduction technique, distinguished by its aqueous-based process. It promptly transitions any amorphous regions in the final product into crystalline form and enhances moisture resistance [29,31].

#### 3.1.2. Spray Drying

Another possible approach involves the use of bottom-up techniques, such as Super Critical Fluid (SCF) drying and spray drying, particularly when dealing with low-dose drugs used in asthma and COPD therapies. It allows the control of the size distribution as well as characteristics of the particles such as their shape and surface morphology. Spray drying enables the production of a dry powder from a liquid starting material (solution or suspension) by quickly drying it with a hot gas. Spray drying can also be executed using excipients (surfactants, lipids, volatile agents, etc.), which creates a variety of particles structures/compositions, hence boosting the pulmonary absorption and bioavailability, as well as regulating the release kinetics and upgrading the dispersion performance [28]. Spray drying is one of the most common methods used in the manufacturing of solid particles. This process finds its popularity in its ability to produce particles with a high shape and diameter uniformity, as opposed to other processes such as crystallization. Spray drying also provides the ability to regulate the morphology of the particles by optimizing parameters such as the temperature of the drying gas, the properties of the precursor (or feed liquid), or the dried droplet local velocity. It is capable of producing, in addition to spherical particles, hollow particles and particles with an increased porosity or roughness. In inhalation, an optimal particle morphology would be a hollow, highly porous structure to achieve a desired deposition. In order to achieve such structure, the precursor needed must contain drug-containing host particles and template particles of known diameter present in a suspension. “Host particles” refers to nanoparticles that encapsulate or carry a drug or active substance, while “template particles” refers to particles with specific sizes that influence the final structure of the spray-dried particles. The precursor or feed liquid is atomized into a chamber, greatly increasing its surface area, to form spherical droplets containing both host and template particles of known concentrations. The droplets are then evaporated by a carrier gas, a warm air stream through a tubular reactor. The template particles are removed either by dissolution in an adequate solvent or through evaporation, leaving the hollow, porous particles in the formulation to be collected in a cyclone [21,29,32]. However, this method possesses disadvantages for stable low dose drugs in a crystalline form since it mainly produces amorphous particles sensitive to moisture and therefore physically less stable [28].

Figure 2 displays α-lactose monohydrate particles which were spray dried by ProCepT 4M8-Trix spray dryer (ProCepT, Zele, Belgium) with the following spray drying process parameters: inlet temperature (Tin): 130 °C; inlet drying gas flow rate: 0.3 m^3^/min and pump speed (feed rate): 10%. The bi-fluid nozzle was used with the nozzle orifice size of 0.6 mm. The picture was taken using a digital microscope (Keyence VHX-970F; Keyence Corp., Osaka, Japan).

Lactose was spray dried from a 10% *w*/*v* lactose solution, to yield micron-sized amorphous lactose particles with a spherical shape (Figure 2). The amorphous state results in a product of a rapid dehydration of lactose particles from the solution during the spray drying process. Lactose is an important excipient DPIs and is commonly used as a filler in tablet manufacturing. As mentioned before, spray drying is a technique that allows particle engineering and relies on specific process parameters and liquid feed compositions to determine particle characteristics. By modifying the morphology and size of powder particles and through co-processing with additional excipients, significant alterations in the aerosol characteristics of dry powders can be achieved [33]. Spray dried lactose compactibility could be improved by investigating the optimum amorphous content [34]. Amorphous lactose is considered to form a binding layer on lactose monohydrate crystalline particles and influences the material’s compactibility [35]. Furthermore, the spherical particle shape contributes to enhanced powder flowability [36]. Commercial spray dried lactose consists of usually of 15–20% of amorphous and 80–85% of crystalline form [37].

Commercially, an example of the use of spray drying would be Vectura Group Ltd.’s dry powder formulation development, which uses this process to control size, shape, surface chemistry, and morphology of particles. This is especially useful for high-dose inhaled products or biologics that require low-energy processing [38].

#### 3.1.3. Spray Freeze Drying (SFD)

Spray freeze drying (SFD) is a possible method to dry thermosensitive compounds, though the powders produced possess poor flow properties and a narrow regulation of the particle size distribution. SFD or cryo-spray drying is a prevalent method which produces particles with an enhanced dispersion behavior [28]. It has notably been used in the production of nanoparticles, particularly porous particles, as done by D’Addio et al. using dried cholesterol and uniform mannitol carriers [39]. SFD is a more intricate and pricey process compared to regular spray drying, therefore it is mostly used for a certain category of pharmaceutical products. The process begins with rapidly freezing a drug-containing solution in a freezing chamber, typically using liquid nitrogen at −196 °C. Droplets can also be injected into a cryogenic medium (spray freezing into liquids, SFL) or into a stream current of −60 °C air running oppositely to the main current (spray freezing into gas, SFG). This step leads to the solidification of the droplets, which will then undergo the second step: lyophilization (solvent sublimation) [29,32]. During the process, parameters can be controlled such as the particle size through the volume of the atomized droplets or the density through the concentration of the solute [21]. A great advantage of SFD processes is the possibility to use them on thermolabile compounds since they are not exposed to high temperatures. Moreover, the engineered particles are characterized by a low density, making them beneficial in inhalation therapy. However, the particles produced also display frailty, which might be a drawback during powder processing in the blending of lactose interactive mixtures [32]. They also possess a high porosity, therefore making them exceedingly voluminous, which means that only a very small amount of drug can be measured in single-dose compartments [28].

#### 3.1.4. Super Critical Fluid (SCF) Drying

SCF technology has been a part of the production of quite a few pharmaceutical applications, notably in nanotechnology. It involves the use of an SCF (carbon dioxide, ethanol, ethylene) above its critical temperature and pressure to create fine drug particles for pulmonary use, notably proteins. It is a distinctive process using a different mechanism to engineer particles in the micro or nanosized range with a comparatively small size distribution [21,29]. SCF processes fall into two categories: one uses the SCF as a solvent in which a solid form material precipitates under SCF decompression, including customary technologies such as the Rapid Expansion of Supercritical Solutions (RESS) or the Particles from a Gas Saturated Solution process (PGSS). The other type involves using the SCF as an antisolvent added to an API mixture in order to decrease the solubility of said API, where Gas Anti-Solvent or Supercritical Anti-Solvent (GAS/SAS) processes are common, along with variations of them [21]. SCF drying is a beneficial process due to its single-step nature, crystal polymorphism control, improved product purity, and eco-friendliness due to the use of carbon dioxide in place of organic solvents and low temperatures [29]. However, it requires intricate and costly equipment. Nevertheless, it proved to be an appealing method for highly sensitive and potent drugs due to the decreased energy and solvent needed. Other advantages of SCF drying include the variety of shapes and morphologies obtainable through this process, such as needled-shaped particles, porous aggregates, etc. [32]. CrystecPharma is a commercial example of a company that uses SCF technology for drug particle engineering. Their focus on supercritical fluid particle design (SCF PD) offers a number of routes to improve solubility and dissolution rate to enhance the bioavailability of poorly water-soluble drugs. The use of online monitoring and computational approaches helps achieving successful solid-state properties manipulation in the creation of pharmaceutical co-crystals and solid dispersions [40].

#### 3.1.5. Electrospinning

Electrospinning is a process involving the use of an electric field to create fine fibers from a solution. It is a versatile technique with significant potential for formulating DPIs. Electrospinning enables the production of nanofibrous materials with tailored morphologies and physicochemical properties, a unique advantage in tailoring the aerodynamic properties of the resulting particles. The process begins with the creation of a polymeric solution, which is then electrostatically drawn into fine fibers, resulting in nanofibers with diameters ranging from nanometers to micrometers. Such nanofibrous structures present a high surface area-to-volume ratio, facilitating efficient drug loading and enhancing the dispersion properties, which is crucial for an optimal inhalation. Moreover, electrospun nanofibers can serve as carriers for a diverse range of therapeutic agents, including antimicrobial agents, proteins, and genes, thereby expanding the therapeutic scope of DPI formulations. However, electrospinning does pose challenges, including difficulties in achieving deposition on diverse substrates (or collectors), a relatively low yield requiring a high working voltage, and complexities associated with large-scale production of nanofibers with specific attributes. Despite these challenges, in comparison to traditional methods such as SFD, electrospinning excels in its precise control over nanofiber morphology and the capacity for a sustained drug release. This makes it an appealing choice to meet the demands of DPI formulations. Moreover, unlike traditional drying methods that may expose thermolabile compounds to high temperatures, electrospinning operates at lower temperatures, preserving the stability of sensitive pharmaceuticals. As this field evolves, the use of electrospun nanofibers in DPI formulations holds great promise for shaping innovative and effective pulmonary drug delivery systems [41,42,43].

#### 3.1.6. Thin Film Freezing (TFF)

TFF emerges as a transformative cryogenic technique due to its ability to engineer dry powder formulations, particularly proteins. Characterized by its ultra-rapid freezing process, TFF provides a meticulous control over powder properties, resulting in particles distinguished by a high surface area and prorosity, amorphous morphology, minimal aggregation, and a submicron size range. This precision makes TFF invaluable in formulating proteins for pulmonary delivery, addressing the customary challenges associated with liquid formulations. Indeed, liquid vaccine suspensions combined with aluminum salts were successfully turned into dry powder without aggregating particles or lowering their immunogenicity. Furthermore, the dry vaccine powder did not agglomerate after repeated freezing and thawing cycles [44]. Moreover, Wang et al. effectively developed particles containing tacrolimus and mannitol using TFF. When the particles were combined with a commercially available DPI, they displayed excellent aerodynamic properties. TFF-processed powders achieved considerably greater pulmonary bioavailability with prolonged lung retention duration in a single-dose dry powder inhalation trial in a rat model, possibly due to their capacity to prevent pulmonary clearance [45]. TFF-produced powders not only maintain functional activity but also exhibit excellent aerosol properties such as in the case for lysozyme, lactate dehydrogenase (LDH) and other proteins, providing enhanced thermostability that could potentially eliminate the need for a cold chain during storage [46,47,48]. An example of the commercialization of this method would be TFF Pharmaceuticals Inc., a clinical-stage biopharmaceutical company which produces dry powders for a targeted delivery to organs such as the lungs. This approach enhances the absorption of poorly water-soluble drugs, improving pharmacokinetic effects and safety profiles. The technology forms “Brittle Matrix Particles” with advantageous attributes such as low bulk density, high surface area, and specific morphology, ensuring structural integrity, functionality, and aerodynamic properties [49].

### 3.2. Excipients

Excipients offer the possibility to improve the non-pharmacologic properties of a formulation. They were formerly deemed to be inert substances but have since proved that they are, on the contrary, useful substances which can be designed to improve the formulation by decreasing the particles adhesion and ameliorate powder dispersion. They can reinforce the physical or chemical stability of a formulation, its mechanical properties, improve the absorption or release of the API, or even act as disintegrant binders, lubricants, filler agents (as mentioned earlier in case of a small drug content), sweeteners, and coloring/identification agents [32,50]. As discussed previously, in terms of pulmonary drug delivery, excipients are specifically required in the formulation to reach the best possible size, thus they are typically found in relatively high amounts in contrast to the API. They supply a bulk mass, hence ameliorating the handling, metering, and dispensing of the drug. The excipient particles are usually produced through milling [32]. Their use may also improve the patient’s compliance, as it can enhance the taste but most importantly the sensation felt by the patient upon inhalation, therefore giving feedback to the patient that the dose was indeed administered [51].

Several components with the potential to improve the pulmonary delivery could also irritate the lungs. Consequently, the excipient options are limited to those easily metabolized or cleared. Among these excipients, lactose (α-lactose monohydrate) is the most frequently used. It has a lengthy use as an excipient in oral formulations and is now present in more than ¾ of the most common marketed DPIs [52,53]. This is due to its multiple advantageous properties such as [54,55]:Physico-chemical stability and compatibility with most low molecular weight drugs;Safe toxicological profile;Availability and affordability;Less hygroscopic than other sugars.

Its highly crystalline nature and good flow properties make it a preferred carrier for APIs in DPIs [55]. However, lactose is unsuitable for diabetic or lactose-intolerant patients since it eventually gets swallowed following its impact on the oropharynx. It can pose risks for individuals with cow’s milk protein allergy (CMPA), as it may contain allergenic milk proteins. Inhalation of these milk proteins may potentially trigger severe allergic reactions [56], and in rare cases, even lead to fatal outcomes [57]. Several instances of lactose in DPIs being contaminated with milk proteins led to allergic reactions, mostly in children (aged 6 to 10) [58,59,60,61] and an adult woman [62]. Lactose is present in 5 (all of which are DPIs) of the 17 inhaled asthma medicines registered by the Food and Drug Administration (FDA). Although anaphylactic reactions from lactose-containing DPIs are mentioned in package inserts, the incidence of such reactions is unknown [60]. Furthermore, a recent study investigated the use of DPIs containing lactose in patients with CMPA. Out of 77 doctors who responded, 45.5% were unaware that DPI leaflets listed CMPA as a contraindication to DPI administration. Additionally, almost all participants were not aware of any systemic allergic reactions in CMPA patients who received lactose-containing DPIs, least of all anaphylactic reactions [63]. Moreover, low-dose APIs in lactose-based adhesive mixtures delivered by a DPI with no clear efficient dispersion principle is a drawback in dry powder inhalation. Despite the addition of magnesium stearate, these mixtures only produce FPFs of up to 40 to 50% of the label claim. The average value among all marketed DPIs today is about 30%, indicating that there is still potential for progress and improvement [64]. Due to these reasons, mannitol, for instance, could represent an alternative to lactose [55,64,65]. Various alternative carrier materials, mainly sugars, have been investigated, leading to varying results depending on the type of inhaler used or even the type of API present in the formulation used. Furthermore, many of the carrier materials that have been tested are not yet authorized for inhalation by the FDA. Another cause for the substitution of lactose in the future is the growing interest in using it as well as lactic acid for other purposes: lactose is sought-after as a sweetening, stabilizing, and moisture-retaining component in food items. Meanwhile, lactic acid is employed as an acidifier in fruit juices and beverages, a preservative, and a flavoring ingredient in pharmaceutical and cosmetic products [64].

It is essential to avoid reducing sugars in formulations comprising amino-group-containing APIs to prevent Maillard reactions and instability [55,66]. To overcome this, non-reducing polysaccharides and non-reducing disaccharides and other sugars are being considered as carriers [54]. Furthermore, a combination of fine carrier particles with the API at an equivalent size range has proven to be a significant factor in the improvement of the formulation performance. Fine lactose particles occupy possible API binding sites on the coarser lactose particles, thus reducing the drug-carrier interactions, as seen with salbutamol sulfate, for example [67]. This improvement is associated with the presence of active sites on the carrier’s surface [67,68,69]. In conclusion, while lactose is common, it may not always be the ideal excipient due to its drawbacks, leading to the development of alternatives such as FDA-approved Mannitol for inhalation use.

Another possibility is to switch from a passive DPI to an active one, achieving dispersion through the use of external energy, or even to exclude the use of a carrier altogether. It is important to note that excipients are not always necessary in a DPI formulation, examples of that are the Oxis Turbohaler^®^ (formoterol) or the Pulmicort Turbohaler^®^ (budesonide).

### 3.3. Types of Particles

Formulation characteristics, including particle size and composition, significantly impact effective pulmonary drug delivery in DPIs. Excipients, produced through techniques like milling, spray drying, spray freeze drying, and supercritical fluid drying, play a vital role in addressing challenges related to particle cohesion and adhesion, allowing for the design of inhalable particles with optimal respirable properties and controlled release potential. This comprehensive approach enhances the success of respiratory drug delivery in DPI formulations and results in a wide variety of available inhalable particles, some of which are illustrated in Table 2.

The different types of particles described in Table 2 undergo specific powder processing techniques to optimize their respirable properties and controlled release potential in DPI formulations. Powder physico-chemical characterization, such as hygroscopicity and crystallinity analysis, is then performed to assess the behavior and performance of these processed particles in DPIs. This thorough characterization ensures the effectiveness and stability of DPI formulations, leading to improved pulmonary drug delivery outcomes.

### 3.4. Powder Processing in DPI Formulations

#### 3.4.1. Powder Physico-Chemical Characterization in DPI Formulations

Hygroscopicity and moisture content are of the outmost importance in DPI formulations. The moisture content (or water content) refers to the percentage of water present in a material, by weight. Hygroscopicity is the tendency of a substance to absorb or adsorb water from its surroundings. It is influenced by both the morphology of the particles in the powder, as well as the crystallinity of the substance. The fluctuations in relative humidity can lead to moisture absorption and loss, therefore resulting in the dissolution and recrystallization of the formulation. This will in turn result in a permanent aggregation of the particles via solid bridge formation [55,137]. In Figure 3, 3 different DPIs were investigated through a comparative analysis of their visual characteristics and behavior under varying environmental conditions. The picture was taken using a digital microscope (Keyence VHX-970F; Lens: Z20:X20; Keyence Corp., Osaka, Japan).

Upon removal from their packaging, capsules from the Braltus DPI displayed a notable aggregation of the powder particles, indicative of a potential moisture-induced cohesion. The aggregation phenomenon was more pronounced following exposure to elevated temperature and humidity conditions, suggesting that the formulation’s susceptibility to moisture was exacerbated under stress conditions. The Onbrez^®^ capsules exhibited a visible powder aggregation, particularly after being stored in a pill dispenser at room temperature for 7 days. This observed alteration became more apparent when the capsules were subjected to elevated temperature and humidity conditions (40 °C and 75% humidity in a stability chamber), hinting at a possible interaction between the capsule material and the internal powder composition. Moreover, the Spiriva^®^ capsules presented an alternative visual cue for stability assessment: while the powder’s condition remained hidden (due to the capsule being opaque), discoloration of capsules stored in the stability chamber indicated a potential sensitivity of the capsule material to the harsh environmental conditions. Collectively, these findings suggest that moisture and temperature fluctuations can influence the stability of DPI capsules, potentially affecting both the dispersion of the powdered formulation and the integrity of the capsule shell. This can further be seen on Figure 4 which displays Scanning Electron Microscope (SEM) pictures of Braltus^®^ capsules powder formulations put in a stability chamber at 40 °C with 75% relative humidity. Each capsule’s content was fixed on a sample holder using double adhesive tape, then gold coating was applied with an Emitech K550X Sputter Coater (Quorum Technologies Ltd., Ashford, UK) for 2 min. Examinations were performed by means of a scanning electron microscope (FEI Inspect S50) at 20.00 kV accelerating voltage. Working distance was between 21 and 22 mm. Original magnification was 300–4000× with an accuracy of ±2%.

Moreover, the hygroscopicity of a material can also affect its adhesive and cohesive properties (especially fine particles in the 1–5 µm size range), and in some (more severe) situations considerably raise the particle size. This increase in size in the formulation before aerosolization (due to hygroscopic growth) would be significantly damaging and would result in the physical or chemical instability of the drug. In DPIs, the physical instability would be more damaging as the irreversible aggregation of the powder would prevent the generation of particles of respirable size. Although hygroscopic dosage forms possess a higher chance of exhibiting physical and chemical instability, interesting approaches to counter that would be to coat the particles with hydrophobic films, or to use excipients influencing hygroscopic properties [55]. Some formulations opt for the blending of the small particles with larger carriers to ameliorate powder flow, which will be discussed later on.

Crystallinity is an important parameter in DPI powder formulations since it influences their hygroscopicity. Solid substances can be present in two states: crystalline, or amorphous. Crystalline materials display clear-cut edges and well-defined faces, as opposed to amorphous materials which tend to have rather curved surfaces. Amorphous materials also do not diffract x-rays and are prone to possess a wide range of melting points. By comparison, crystalline materials show well-defined x-ray diffraction patterns and display precise, steep melting points. Depending on their manufacturing methods or storage conditions, DPI powders can be present in either of these states. The impact of amorphous form on stability is a critical consideration for DPI formulations. Extensive studies of the physical stability of amorphous pharmaceuticals, such as those discussed by Shetty et al. [31], have demonstrated that amorphous forms of drugs may exhibit instability over time unless they are in a solid glassy state. This instability arises due to the potential for amorphous materials to morph from a glassy state to a rubbery state when exposed to rising relative humidity. This change in molecular mobility can lead to crystallization, significantly affecting the stability of the final dosage form, leading to altered therapeutic effects. Understanding the solid-state behavior of DPI powders during manufacturing and storage is paramount for ensuring product quality and maintaining therapeutic effectiveness. Characterization techniques, as detailed by Shetty et al., play a crucial role in detecting the solid-state instability of dry powders. These techniques include powder X-ray diffraction (PXRD), spectroscopic methods like Raman and Fourier-transform infrared spectroscopy (FTIR), and thermal analysis methods such as Differential Scanning Calorimetry (DSC), dynamic vapor sorption (DVS), and thermogravimetric analysis (TGA). These tools provide valuable insights into phase changes and transformations under various environmental conditions. It is also worth noting that surface diffusion in amorphous molecular materials plays a significant role in shaping the solid-state properties of powders, especially at temperatures below the glass transition temperature [29].

#### 3.4.2. Formulation Characteristics of DPIs

Since the optimal particle size for a suitable deposition in the lungs is around 3 µm, powder formulations intended for pulmonary delivery customarily consist of micronized drug particles in the 1–5 µm particle size range. Because of this size range, these particles are cohesive, adhesive, making them prone to agglomeration and to adhering to the surfaces of the device (mainly through van der Waals forces) [138]. Moreover, they show poor flow properties, and are usually electrostatic, which leads to a problematic processing, device metering, and air stream dispersion. This is why, in the case of DPIs, they are commonly blended with an inactive excipient of greater size (40 µm) [54,139]. Examples of these excipients are lactose, mannitol, glucose, sorbitol, sucrose, and trehalose [53,139]. However, more recently, the development of particle engineering has led to the possibility of producing dry powder formulations using only pure active ingredient, as seen with rifapentine by Chan et al. [52,140]. Additionally, to overcome the challenges posed by the cohesive and adhesive nature of micronized particles commonly used in pulmonary delivery, an alternative approach has been explored. Non-carrier-based formulations, which involve using pure active ingredients without the need for additional carriers or excipients, have emerged as a promising solution to enhance powder flow properties and streamline device processing, metering, and air stream dispersion. In order to surmount the difficulties the micronized particles present, four main types of formulations are developed: non-carrier-based, carrier-based, large porous particles, and agglomerates [54,138,141].

Non-carrier-based formulations (or carrier-free formulations) employ advanced techniques to create fine, inherently flowable, and dispersible drug particles. These innovative formulations eliminate the need for a carrier material and hold promise for enhancing drug delivery efficiency and improving the patient experience in inhalation therapy. This novel approach aims to enhance the uniformity and dispersal properties of inhaled powders, leading to improved effectiveness of respiratory medications. Moreover, Varun et al. found that carrier-free formulations are the preferred choice when aiming for high drug doses through DPIs [142]. Notable examples of carrier-free particles currently in use include spheroids formed by combining micronized budesonide (Pulmicort^®^) and porous particles known as PulmoSpheres^TM^ [143]. Wong et al. developed, through spray drying, a carrier-free DPI formulation which consists of a 1:1 cocrystal of favipiravir and theophylline. It presents a potential substitute treatment approach for patients with both influenza infections and asthma/COPD, exhibiting desirable characteristics for pulmonary delivery, without requiring a carrier. It exhibited an improved dissolution rate and showed a favorable in vitro cytotoxicity profile [144]. In addition, research indicates that spray-dried non-carrier-based DPI particles yielded a higher FPF percentage [142]. Moreover, ongoing research is actively exploring diverse types of carrier-free particles [143,145]. The primary obstacle lies in overcoming the significant cohesive and adhesive properties of the micronized particles. influenced by physicochemical attributes such as crystallinity, surface free energy, size, density, and shape [145]. Indeed, according to Azari et al., morphology appears to be the most essential concern to avoid drug aggregation during the aerosolization stage in the carrier-free DPI formulation of spray dried Ketotifen fumarate [146]. While non-carrier-based formulations offer potential solutions to enhance drug delivery efficiency and improve patient experience in inhalation therapy, carrier-based formulations remain the most prevailing method for small-sized particles suitable for inhalation, using excipients to improve dose reproducibility and facilitate drug dispersion.

Carrier-based formulation (or adhesive mixture) is the most prevailing formulation method for small sized particles suitable for inhalation. It comprises two elements: the API and an excipient acting as its carrier. The API is usually mixed with larger, coarser particles to ameliorate the dose reproducibility. While usually 40 µm, the carriers can also be of a greater size (500–200 µm) [20,147]. The carriers should meet specific requirements such as providing bulk, ensuring flowability, decreasing particle agglomeration, facilitating powder handling (by increasing the formulation volume), and aiding the dispersion of micronized drugs [54,148]. The two components are blended together, and the drug particles (smaller) adhere to the surface of the carrier particles (larger), thus forming an adhesive mixture. In the mixture, the interparticulate forces binding the drug to the carrier are required to be strong enough to yield a stable, uniform, homogenous blend despite the considerable size and concentration differences between the two components, yet weak enough to enable the segregation of one from the other. Micronized particles commonly adhere to a solid surface through physical forces (such as the van der Waals force), interlocking forces, electrostatic force, and capillary force [20].

In DPIs, the powder formulation (with a size of 100–150 µm) is stored in a capsule in its aggregated form. A deep and vigorous inspiration through the inhaling device is required to de-agglomerate the powder formulation into respirable particles of 1–5 µm. This process is a fundamental requirement for DPIs capsules contents [149,150]. The drug delivery occurs through three different steps [54]: the detachment of the API from its carrier, their dispersion in the airflow, and the deposition in the pulmonary system (Figure 5).

During the detachment step, the carrier may remain in the inhaling device itself or eventually deposit in the oropharyngeal region. The excipients used as carriers are considered to have a somehow restricted loading capacity, which makes adhesive mixtures more appropriate for low drug doses. The use of low dosed drugs as DPIs is mainly directed to respiratory diseases such as asthma or COPD. In the respiratory drug delivery, a drug’s dose is considered low when it is present in the µm range (<1 mg) [152]. Low drug doses can vary from 6 µg of formoterol fumarate in the Oxis^®^ DPI to 500 µg of fluticasone propionate in Flixotide^®^ or Seretide^®^ DPIs [153].

Common drug/carrier ratios are 1:67.5 or 1:99 [52,54]. The quantity of API which can be formulated is limited by the content uniformity and stability requirements [152]. Depending on the nature of the excipient used as carrier in the formulation, the drug quantity limit is set to 5–10% [154]. There are several types of adhesive mixtures which differ by their mixing conditions, drug content, and the type and size distribution of the carrier. The alteration of these characteristics changes features such as the flow properties, formulation dispersion during inhalation, and thus the dose ultimately delivered. In views of the amount of carrier being considerably higher than the amount of drug, its physico-chemical characteristics are of the outmost importance in the formulation. An uncompleted drug/carrier detachment would result in a poor lung deposition and is considered the main reason for the decreased effectiveness of many DPIs. The goal would be to improve the detachment without increasing the inhaler’s resistance to airflow above the patient’s aptitude, which has been achieved in more recent devices [28,55,153]. While carrier-based formulations are effective, they may present challenges for patients with specific respiratory conditions who struggle with inadequate inspiratory flow rates or have sensitivity to the carrier material. Among carrier-based formulations, large porous particles have gained attention as a specialized type of carrier.

Large sized porous particles (or hollow particles), such as PulmoSpheres^TM^ particles (produced by emulsion-based spray drying), represent a promising category for pulmonary drug delivery. They offer several advantages over small non-porous particles with higher density [16,155,156]. In addition to their density, they possess a small aerodynamic diameter but also a relatively large geometric diameter which allows them to aggregate less and disperse more easily [54]. Their development was also motivated by their ability to decrease phagocytosis of the particles in the alveolar region. The deposited particles do not undergo macrophages clearance, making remain longer in the alveoli, which would be beneficial for a slow release inhalation formulation [138]. The pulmonary delivery of a formulation containing particles with a geometric particle size up to 20 µm is an important example to illustrate this method [54,157]. Nonetheless, the lungs small airways (peripheral airways of less than 2 mm internal diameter) may induce deposition of the particles by interception before reaching the deeper lung. To address similar particle size-related challenges, controlled agglomeration through spheronization has been explored as an alternative method for pulmonary drug delivery.

Agglomerates are also a possibility to overcome the particle size related issues. Controlled agglomeration can be achieved by spheronization into soft pellets which are strong enough to be handled but also weak enough to de-aggregate into particles of optimal size for respiration, since the loose agglomerates are bound together by weak interactions. This formulation process would be suitable for the delivery of drugs with a high dose in the mg range [54]. An example of that was performed by de Boer et al., where it was demonstrated that high particle fractions were obtained by de-agglomeration, without any particular particle engineering process to decrease interparticulate forces [158]. Using the Twincer^®^ inhaler, they showed that up to 25 mg of pure powdered drug could efficiently be de-agglomerated, with a possibility to be increased to 50 mg [141]. The primary hindrances of this process are the strong requirements on the production process and the precision of the inhaling device’s metering [28,54].

### 3.5. DPI Formulations Testing

All in all, dry powder formulations call for a specific technical design and manufacture. The formulation should be processed in a way which makes the discharge of the device formulation-containing compartments and the duplicability of the dose measured undemanding. The formulation should also be satisfactory and suitable so that the API reaches and deposits on the desired target area with an appropriate flow rate. As a result, the drug and carriers should be present in an adequate aerodynamic size distribution, and then be dispersed properly in the airstream inhaled by the patient. Other requirements for the formulation are satisfactory flow properties but also the regulation of the interparticulate forces. As seen before, different types of formulations are elaborated to achieve the right conditions, and special particle engineering procedures might be used. Each of these different methods can be considered the most adequate depending on the drug dose, properties, deliverance objectives, but also on the type of inhaler used [28]. Since the processes involved are rather complex, thorough testing is necessary to ensure the efficiency, quality, and safety of the formulation through general and supplementary tests. The most common ones are presented in Table 3.

It is important to note that specific tests performed on DPI formulations can vary: in addition to the common tests listed, there may be other tests conducted to assess the performance, quality, and safety of DPI formulations such as stability testing or device-related testing.

Particle size determination is a crucial step in the formulation of DPIs, as stated previously. Cascade impactors (listed in the Pharmacopoeia) are the most common devices used for the in vitro study and measurement of the particle size distribution and other parameters of a dry powder formulation. They allow a direct measurement of the aerodynamic particle size and the determination of the drug mass across different size ranges, while excluding any disruption from the excipients

Particle size can also be determined by optical methods. The most commonly used as an alternative to cascade impactors is laser diffraction, but light scattering, laser Doppler, and time-of-flight can also constitute possible substitutes [161]. Although particle size measurements can be completed by laser diffraction, aerodynamic diameters cannot be obtained with laser diffraction as opposed to using cascade impactors. Next Generation Impactors (NGIs) also allow the determination of parameters such as the FPF and other size fractions, while laser diffraction and other techniques do not provide any differentiation and simply measure the overall particle size distribution in the sample. Nonetheless, laser diffraction is a method which has been used since the 1980s [162] for nebulized drug solutions particle size measurements and is considered to be a fast, highly accurate, flow-rate-independent method thus constituting a great alternative to NGIs.

Dissolution testing informs us about a formulation’s in vitro drug release and absorption behavior by assessing its ability to penetrate a solvent medium based on their affinity [161]. For non-parenteral formulations, understanding their dissolution profile is essential for evaluating bioavailability. The dissolution behavior of these dosage forms can be influenced by the drug’s solubility, dose, particle properties, but also the formulation properties, and the epithelial lining fluid (ELF) composition which changes along the respiratory tract. In the lungs, drug absorption primarily occurs in the small bronchioles and alveoli, where drug dissolution is the most significant. Following the inhalation, the drug will dissolve in the ELF (composed of a surfactant layer and an aqueous phase) along the respiratory tract [163]. As mentioned, the ELF varies along the respiratory tract. Depending on the region, it will differ in composition, thickness and volume. The trachea, bronchi, and bronchioles are coated with a thick mucus gel (about 3–23 µm) whereas the alveolar region is layered with a very thin film (about 0.07 µm). As the drug travels down the respiratory tract, the lining fluid progressively gets thinner, resulting in physiological disparities, and therefore rendering it difficult to establish the residence time of the particles through simulating lung conditions [164]. Following inhalation, the particles which penetrate the non-ciliated part of the respiratory tract will dissolve in the ELF and constitute the only part of the dose administered accessible for absorption through the alveolar membrane [161].

Although a few new dissolution and permeability testing methods have been established, none have emerged as the standard method of choice [161,164,165,166]. An example of these tests is the paddle over disc dissolution setup, which investigates the in vitro dissolution rate of inhalation dosage forms [167]. Another possible technique is the flow-through cell apparatus which evaluates the dissolution profile of poorly soluble glucocorticoids inhaled formulations [168]. As an alternative, the Franz diffusion cell apparatus can be used to investigate the dissolution profile of pulmonary formulations, which has proven to be the most promising out of the three methods mentioned above [161,169].

Adequate testing of the dissolution of an inhaled formulation is an intricate process notably due to the physiology of the lungs such as the small amount of fluid they possess, but also because the API should be successfully separated from the excipients before testing [161]. Unlike the alveolar region, the tracheobronchial part of the lungs is coated by a viscoelastic blend composed of proteins, glycoproteins, and lipids. Nonetheless, this mucus’ composition can change in case of ailments such as infections [170]. In vitro testing requires an accurate simulation of in vivo conditions, for pulmonary testing the most accurate option is the use of biological simulated lung fluid (SLF) as a dissolution media. SLF was developed in 1979 by Moss [171]. In a study made by Hassoun et al., a biorelevant SLF was successfully designed, with a precise composition and characteristics as well as usage and storage directions. It showed physico-chemical properties comparable to those of the lungs’ lining fluid and can be used for in vitro investigations of the dissolution of inhaled formulations, among other uses [172]. Although SLF supplies a great insight into in vivo mechanisms, the creation of an accurate, standardized lung dissolution testing method for inhalation remains a complicated endeavor due to the lungs’ characteristics.

Physico-chemical properties characterization also plays a crucial role in the stability of a powder formulation. When it comes to physical properties, the surface of the particle is of great relevance to assess the particle stability. Imaging techniques such as SEM and AFM (Atomic Force Microscopy) are great tools to show the changes of particle morphology during certain storage conditions. SEM scans the surface of the particle with a beam of electrons instead of light. It informs us about the particles’ surface topography, crystalline structure, chemical composition, and electrical behavior [29,173]. Figure 6 showcases SEM pictures (with the same parameters as Figure 4) of dry powder formulations from 3 inhalers (Spiriva^®^, Onbrez^®^, and Braltus^®^), highlighting their structure and features.

In regards to the chemical properties of DPI formulations, the surface chemistry helps us identify the interparticulate forces as well as the formulation’s aerosolization capacity. Techniques such as EDX (Energy Dispersive X-ray spectrometry) possess spatial resolutions in the nanometer range. EDX is used for the chemical characterization of a dry powder sample through elemental analysis. It relies on the production of distinctive x-rays which help determine the chemical composition of the sample. It can be used to characterize the surface composition of API or excipient-coated materials for DPI formulations. EDX also shows some drawbacks such as a low detection limit and an x-rays penetration depth in the micrometer range [29,174].

## 4. Design and Performance Considerations for Inhaler Devices

### 4.1. Performance Assessment of Inhalers

The performance assessment of DPIs can be achieved through the patient’s inspiratory flow and the inhalation device’s internally generated turbulence. DPIs are breath-actuated devices, therefore the patient has to supply sufficient turbulent inspiratory forces to break down the powder formulation into fine particles of < 5 µm for adequate lung deposition. In order to achieve an optimal use of the inhaler, an adequate patient’s inspiratory flow, as well as the turbulence formed by the intrinsic resistance of the device (influenced by the design of the inhaler) are important factors. Three levels of resistance are available: low (e.g., Breezhaler^®^), medium (e.g., Ellipta^®^) and high (e.g., Handihaler^®^) resistances. The incorrect manipulation of the device affects the system, which can be shown by the duplicability of the dose by the inhaler at different flow rates. A high patient’s inspiratory flow rate increases the inhaler’s performance by increasing the drug dose inhaled by the patient [175,176]. The inspiratory flow passing through the device should be approximately 60 L/min to achieve deaggregation of the powder. The airflow resistance specific for a device can be determined by the flow rate and the pressure drop using Ohm’s law, as described in Equation (5). In children and elderly patients in general, a higher airflow resistance is more problematic when it comes to using the inhaler adequately with an optimal flow rate, thus the airflow resistance constitutes a significant parameter to consider.
(5)$$R=\frac{\Delta P}{Q},$$
where:*R* = resistanceΔP = pressure dropQ = flow rate

### 4.2. Patient Compliance and Device Optimization

Assessing the performance of an inhalation device also means assessing the patient compliance associated with it. Although the painless pulmonary delivery increases the patients’ compliance as opposed to the parenteral route for example, it is important and necessary to inform and instruct them about the proper use of inhalation devices. Many patients mishandle the devices, mostly due to a faulty coordination of the patient’s inhalation with the actuation of the device. In a study made by Janežič et al., 70%of the patients made at least one mistake in their inhalation technique [177]. In another study conducted by Arora et al., it was observed that 82.3% of the patients studied made at least one mistake while using their inhaler. The three different types of inhalers were used in the study, resulting in MDIs users being the group who mistakenly used their device the most (94.3%). In second were DPIs-using patients (82.3%), followed by MDIs + spacer users (78%), and finally the nebulizers group (70%) [178]. An additional study carried out by Molimard et al. in primary care in France resulted in 76% of over 3800 outpatients making at least one mistake while using a pressurized metered dose inhaler (pMDI) [179]. In order to rectify this, the inhalation device should be optimized and should respect fundamental requirements such as its user-friendliness, which can be checked through implemented feedback mechanisms as it is the case for the Novolizer^®^ for instance, which incorporates feedback signals showing the patient that an adequate amount of drug has been released from the device optically through a color change (green to red) in a control window, but also audibly by means of a “click” sound. The presence of a dose counter also helps keep track of the therapy and patience compliance [180].

### 4.3. Impact of Storage Conditions on DPI Capsules and Blisters

Capsules not only contain the formulation of DPIs but also shield it from potential changes such as moisture absorption. This protective function helps maintain the stability and the effectiveness of the enclosed medication. In order to investigate the potential changes in DPI formulations, we investigated the mass of three different inhaler capsules after they have been stored at room temperature outside their blister/container packaging. The goal was to simulate changes when the capsules are stored in environments such as pill dispensers, which is not a rare practice among some patients. To conduct the experiments, 10 capsules from 3 inhalers (Spiriva^®^, Onbrez^®^, and Braltus^®^) were selected. The initial mass of these capsules was measured on day 0, immediately after removal from their packaging. Subsequently, the same capsules were weighed after being stored at room temperature for 7 days in a pill dispenser. Figure 7 shows the average mass of each type of capsule at day 0 and day 7.

The experiment revealed a change in the mass of the capsules over the 7-day storage period: the Spiriva^®^ capsules showed the highest change, with an increase of approximately 5.1%. Similarly, Braltus^®^ capsules showed an increase of approximately 5.0%. And finally, Onbrez^®^ capsules showed an increase of about 0.5%. The observed increase in mass for all three DPIs indicates the potential influence of external factors on the capsules, such as moisture absorption. Formulations, especially the ones with hygroscopic properties, can interact with the environment’s moisture. The mass increase may have implications for the uniformity of the drug delivery, which is relevant for optimal therapeutic outcomes.

Building upon the insights we gained from the initial experiment conducted at room temperature, we further explored the effects of altered storage conditions on the capsules. The second experiment involved placing the capsules in a controlled stability chamber maintained at 40 °C and 75% relative humidity to simulate more extreme conditions. As seen on Figure 8, the results showed a similar trend to the previous experiment: both Spiriva^®^ and Braltus^®^ had an approximately 6.9% increase, while Onbrez demonstrated an increase of about 0.7%.

Once again, these observations suggest that the capsules are responsive to changes in their environment. These findings underline the importance of considering storage conditions and their effect on delivery consistency.

Likewise, the water content within the capsules from each inhaler was investigated over different periods of time and storage conditions. Measurements were completed straight from the packaging, after 7 days at room temperature in a pill dispenser, and after 7 days in a stability chamber with the same parameters as before, as shown on Figure 9.

The results indicate changes in the water content for the different inhalers: Spiriva^®^ showed a decrease in the stability chamber (−14.8%) and a very slight increase at room temperature conditions (0.0132%). This contrasting behavior suggests that the formulation’s sensitivity to moisture is more pronounced under controlled harsh conditions. Braltus^®^ on the other hand displayed a decrease in water content for both conditions, −6.1% at room temperature and −2.8% in the stability chamber, Alternatively, Onbrez^®^ showed significant alterations in the water content, with an increase in both conditions, 17.8% at room temperature and 4.3% in the stability chamber. This suggests that the formulation exhibits a notable tendency to absorb moisture from its environment. The change in water content could be attributed to the capsule/formulation’s ability to interact with moisture from the environment, similar to what was observed in the mass experiments. The decrease in water content on the other hand might be due to moisture loss over time due to the elevated temperature of the stability chamber or possible interactions with other components of the formulation. Moreover, the capsule material (discussed later) may also play a role in these changes.

Furthermore, for a successful aerosol formation and pulmonary delivery, a dispersion device, i.e., an inhaler, needs to be used with the formulation. It needs to be developed in a way which guarantees a reproducible dose each time it is used and the delivery of particles with an adequate size distribution. With DPIs, every device regardless of the type generally consists of four main components: a dosing system (containing or measuring a single dose), the powder formulation, a powder de-agglomerating system, and the mouthpiece. Secondary components may be added to optimize the device for various reasons, such as ease of use, patient feedback, or even moisture protection. The dosing system commonly consists of single doses of the formulation already weight and placed in a capsule or blister. It usually works by piercing or opening the capsule/blister through a manual maneuver applied by the patient, causing the release of the capsule/blister powder and its dispersion in the air stream to occur concurrently. In some cases (e.g., Spiromax^®^), certain devices contain mechanisms which apply further de-agglomeration processes to raise the FPF. Otherwise, multi-dose reservoirs also constitute a design possibility, with certain requirements for flowability and homogeneity of the powder formulation. Single doses are separated and placed into compartments or orifices, for instance, on a disk (e.g., Turbohaler^®^). The orifices are filled from the powder bulk reservoir mostly by gravity, which necessitates the patient to keep the device in a vertical position. In specific instances, the metering is forced by the use of compressed air pushed through the powder bulk reservoir [138].

Buttini et al. emphasized the role and significance of capsules in inhalation therapies. The review showed that ideal capsules for inhalation need to fulfill several key criteria, including the ability to be easily punctured or cut without shedding excessive shell particles. When punctured, the flaps generated should remain attached, open, and not hinder the powder discharge. Additionally, powders should empty from the capsule with minimal retention and interaction between the shell and the fill material. These characteristics are influenced by the capsule’s material, moisture content, and the level of internal/external lubricant. Importantly, reducing the shell’s moisture content should not lead to capsule brittleness. When it comes to capsule containers, two types of capsule polymers are available: hard gelatin capsules and hydroxypropyl methylcellulose (HPMC) capsules. The production process for these capsules involves specific techniques. For inhalation therapy, capsule dissolution or disintegration tests are not critical, but other attributes such as moisture diffusion, physical and mechanical performance, and lubricant content are important. Gelatin capsules have been widely used in DPIs for over 30 years but are made from the hydrolysis of collagen and are susceptible to brittleness when their moisture content decreases. This can lead to issues with puncturing and may result in patients inhaling small fragments. Modified capsules containing plasticizers have been developed to address this problem, but challenges with releasing shell fragments persist. To address this, HPMC capsules were introduced, offering better stability and aerosolization properties. They have a lower moisture content (4.5–6.5%) compared to gelatin capsules (13–16%) and do not become brittle at low humidity. HPMC capsules are chemically inert and composed of vegetable sources, making them suitable for vegetarians and avoiding ethical concerns. Manufacturers have developed different processes to produce them, including thermal gelling and cold gelling methods. HPMC capsules generally have slightly higher moisture content than gelatin capsules, though extra dry capsules were introduced to improve product stability and reduce the need for post-filling drying. Moisture content is critical for capsule stability and mechanical performance. In this regard, HPMC capsules are more effective in reducing moisture content, and they perform better in puncturing tests compared to gelatin capsules. Furthermore, lubricants are essential for capsule production, and the level of lubricant influences drug deposition and fine particle dose. However, the overall choice between gelatin and HPMC capsules depends on how they interact with the formulation and the specific needs of the inhalation therapy for which they are used. Indeed, proper capsule characteristics are crucial for successful inhalation therapies using DPIs [7]. HPMC capsules also demonstrated greater stability and aerosolization properties compared to gelatin (GEL) and gelatin capsules with polyethylene glycol (PEG) modification (GEL-PEG): Benke et al. studied the influence of different capsule types on the stability and aerodynamic properties of Ciprofloxacin-containing DPI formulations. Over time, GEL and GEL-PEG capsules exhibited decreased stability, leading to irregularly shaped holes and reduced aerosolization efficiency. HPMC capsules, on the other hand, demonstrated better stability and maintained regular hole shape, making them a more favorable choice for DPI formulations [181].

Moreover, a recent study performed by Ding et al. discussed the impact of different inhalation-grade capsules on aerosol performance for both carrier-based and carrier-free formulations. The study compared various capsule types and their effect on aerosol performance parameters such as the emitted fraction (EF), FPF, and MMAD. The findings showed that the choice of capsule type plays a significant role in aerosol performance, especially for carrier-free formulations. Among the capsules evaluated, Embocaps^®^ VG capsules demonstrated the best aerosol performance for the carrier-free formulation, possibly due to their exceptional hardness. For the carrier-based formulation, although no statistically significant differences in aerosol performance were observed, variability in performance was noted, which might be linked to formulation differences, environmental factors, and capsule properties. Additionally, the study reviewed the influence of capsule piercing and the detachment of capsule flaps on aerosol performance. The hardness of the capsules seemed to impact the degree of flap detachment and subsequent aerosol performance. This study emphasizes the importance of carefully selecting suitable capsules and highlights the importance of the relationship between capsule properties and formulation differences, advocating for further investigation of different capsule types to optimize the performance of novel inhaled formulations [182].

Despite capsules being widely used as a means of dose administration in a significant number of DPIs, pharmaceutical companies explored alternative dosing options for medications, one of which being blisters. A standard blister pack is composed of a base material containing one or more cavities where the drug is inserted, along with a lidding film that securely seals the base and encloses the cavity. Blister packs are prevalent in the pharmaceutical industry for packaging unit doses of powders, pills, and capsules. For particularly moisture-sensitive materials, blister-based DPIs are the preferred choice. These devices feature a series of aluminum blisters arranged in a ring, with each blister containing a single pre-measured dose of drug powder. To keep track of the doses administered, the device is equipped with a dose counter. When the inhaler is activated, it pierces the blister, and the airflow generated during inhalation facilitates the release of the powder from the blister. It is crucial to ensure that the chosen blister material preserves the drug dose from any physical or chemical alterations. Simultaneously, the blister should possess suitable rigidity and mechanical properties to facilitate effortless piercing and dose release. DPI blister packs are usually made of polypropylene, polyvinyl chloride, polystyrene, and aluminum [183]. For instance, in the Exubera^®^ DPI, each blister card consists of six perforated unit dose blisters made of polyvinyl chloride (PVC)/Aluminum. These blister cards are then placed in a thermoformed tray made of clear plastic (polyethylene terephthalate—PET), and each tray contains five blister cards along with a desiccant. The entire setup is covered with a clear plastic (PET) lid. Finally, the tray, along with the desiccant, is sealed in a foil laminate pouch to ensure protection [184]. Examples of blisters-based DPIs include: Diskus™ (salmeterol and fluticasone combination), Diskhaler^®^ (zanamivir), Ellipta^®^ (fluticasone furoate, umeclidinium, or vilanterol trifenatate), Acu–Breathe™ (fluticasone propionate), Microdose^®^ (atropine), and Puffhaler^®^ (measles vaccine) [183].

## 5. Inhalation Delivery Systems: Dry Powder Inhalers

There are three main categories when it comes to marketed inhalers: nebulizers, MDIs and DPIs. Their categorization is completed according to the physical states of the dispersed phases and continuous media, with further differentiation within each category based on the mechanisms of dispersion, design, and metering [55].

### 5.1. Dry Powder Inhalers

Dry powder inhalers (DPIs) are small portable devices considered simple to use in that minimal patient input is needed between actuation and inhalation. They are also propellant free inhalers, enclosing a micronized powder formulation, a dosing or dose measuring system, and a mouthpiece. The solid active ingredient is blended with a powder mix which is fluidized with the inhalation from the patient [55].

#### 5.1.1. Advantages of DPIs

Both the drug formulation and the device itself have an influence of the aerosolization performance of DPIs, and their improvement can ameliorate both aerosolization and patient adherence. The inspiratory flow of the patient is the main driving force used to breakdown and deliver respirable particles to the respiratory system. DPIs were developed with the intention of providing alternative options to pMDIs, with the aim of minimizing the release of ozone-depleting cholorofluorocarbons (CFCs) and greenhouse gases (hydrofluoroalkanes—HFAs) that are utilized as propellants, and to simplify the delivery of macromolecules and biotechnology products. Indeed, dry powder pulmonary formulations are suitable for the delivery of biologics including proteins, nucleic acids, phages, but also genes, peptides, virus, and monoclonal antibodies. There is also an increased interest in inhaled vaccines, especially in the aftermath of the COVID-19 outbreak, which is expected to expand research efforts in inhaled biotherapeutics, particularly mRNA- and protein-based vaccines against pulmonary infections. However, the development of these inhalable dry powder biologics represents challenges in their formulation and manufacture. Furthermore, their stability, dispersibility and excipient selection are of critical importance, making long-term studies required to ensure their safe and effective delivery, though a substantial advancement was made in engineering particles for inhalation over the past three decades to enable the effective delivery of biologics [33,185].

Moreover, the development of DPIs was able to effectively tackle various drawbacks related to the formulation as well as devices themselves faced by pMDIs. DPIs are considered to be more user-friendly, stable, and efficient systems [55]. Just like pMDIs, DPIs benefit from a short treatment time and are small and portable. However, another significant advantage they have is a propellant-free formulation and their user-friendliness (no patient coordination needed). An additional advantage of DPIs over nebulizers is the reduced risk of contamination due to the dry environment for both the device and its formulation [186,187]. Despite that, the dependence on the patient’s inspiratory flow as well as the particle aggregation due to humidity constitute drawbacks for this type of inhalers. This is a relevant issue considering that in a study, 42% of the patients stored their inhaler in their bathroom and 21% in their pocket/handbag, with only 4% storing it properly according to the given instructions [188].

#### 5.1.2. Innovations in DPI Technology

One way to overcome these difficulties is the use of smart devices, which are developed with electronic monitoring systems allowing their connection to other devices or to an internet network. Their objective is to lessen the patient and device errors and can track the patient’s adherence to the therapy. They aim to enhance the treatment’s outcome through reminders prompting the patient to take the dose, or by reminding them of the correct use of the inhaler step by step by displaying it on the device’s screen. An example of “originally integrated” devices (as opposed to “add-on” devices) is the 3M™ Intelligent Inhaler by 3M™ Drug Delivery Systems [189]. Moreover, in an attempt to achieve an inspiratory flow-independent inhaler performance, inhalers that employ outside energy to disperse powder have been developed. For this purpose, three types of energy have been exploited: compressed air (e.g., Exubera^®^—Pfizer), electrical energy (e.g., Taper^®^—3M™), and thermal energy (Staccato One Breath Technology™—Alexza) [64].

#### 5.1.3. Classifications of DPIs

DPIs can be classified according to different criteria. The first is the number of doses per device: DPIs can be single-unit dose (disposable or reusable), multi-unit dose, or multi-dose reservoir.

Single-unit dose DPIs are either disposable or reusable but for both, the dose is available in an individual capsule. A single dose delivery is achieved by the patient placing the capsule inside the inhaler before each use. The capsule shell remains in the device while its content flows through it and into the patient’s respiratory tract. Examples of single-unit dose inhalers are the Handihaler^®^, or the Manta’s SOLO™ inhaler (which is disposable). Reusable DPIs possess several advantages, one of which is their low cost of therapy since the inhaler can be reused numerous times. They are also convenient for high-dose antibiotics since several high doses are required and a multi-dose DPI bearing them all would be too large and bulky. An example of that is the TOBI^®^ Podhaler^®^ with the aminoglycoside antibiotic tobramycin. Single-unit dose DPIs on the other hand are convenient in the delivery of single dose medication such as vaccination or in therapies requiring infrequent dosing. The frequency of their use requires them to bear a low cost and be easy to use.

Multi-unit dose DPIs contain previously metered, sealed, packaged doses on disks or blister packs, therefore making the device loaded beforehand and carrying several doses at the same time. This type has a significant advantage in that the pre-measured doses ensure that the accuracy of dose metering is not affected by the patient’s inhaler handling. However, some devices’ drawback is their limited number of doses, with the Diskhaler^®^ only containing four to eight doses. Examples of multi-unit dose DPI are Diskus^®^, Seretide^®^, or Ellipta^®^.

Multi-dose reservoir DPIs include a reservoir which contains the powder formulation in bulk, discharging up to 200 doses [64]. Upon actuation, usually via a twisting of the base of the device, an individual dose is metered, disintegrated, and delivered to the patient. Unlike most capsule and blister inhalers, several of them feature a more effective dispersion principle that results in higher FPFs at lower flow rates. The inhaler must be held in the proper position for accurate dose metering into the cavity, which is completed by gravity and requires good flow properties of the powder formulation. Almost all the devices in this category comprise adhesive mixtures formulations, with the exception of the Turbohaler^®^ and Twisthaler^®^ which contain soft pellets accompanied by a low quantity of lactose [64]. Examples of multi-dose reservoir DPI are the Symbicort Turbohaler^®^ or the Orion Easyhaler^®^.

A second criterion to classify DPIs is their powder dispersion mechanism. They can be either actively or passively actuated.

Active DPIs possess a built-in energy source independent of the patient’s inspiratory flow which aerosolizes the powder formulation. Energy sources examples are compressed air, electrical vibration, or heat [186]. The Spiromax^®^ inhaler is an example of actively actuated DPIs: it is designed with a battery as the energy source. Following actuation, the powder is aerosolized and the device’s impeller causes impaction and consequently powder dispersion. A weak inspiration flow from the patient is sufficient to activate the device [139]. However, active devices’ elevated price and reduced portability remain considerable disadvantages when compared to passive devices. Additionally, even though active devices are independent from it, the patient’s inhalation pattern can still influence throat deposition since the aerosol undergoes inertial impaction [186].

Passive DPIs are the dominant category of marketed DPIs [32]. They are actuated by the patient’s inspiration which produces an airflow powerful enough to de-aggregate the powdered formulation. As the patient inhales and actuates the DPI, the airflow generates shear and turbulence through the device. This inserts air into the powder bed, causing the static powder blend to become fluidized and penetrate the patient’s respiratory tract. In the airways, drug particles separate from carrier particles and penetrate deeply, while larger carrier particles are cleared as they impact in the oropharynx. Consequently, the variability of the patient’s inspiratory airflow determines the deposition of the drug into the lungs [55]. DPIs have been customary in powder aerosolization but their airflow dependency put them at a disadvantage compared to active devices [186]. The Rotahaler^®^ and the Spinhaler^®^ devices are one of the first passively actuated DPIs available on the market. After manually loading the device, the patient actuates it (hence perforating the capsule) and takes a deep breath to provide the necessary force for the impeller in the device to whirl the powder released from the capsule. One drawback of passive DPIs is the contrast among different patients’ inspiratory forces (according to age, disease stage, etc.) but also the absence of uniformity among inspiratory flows from the same patient, which affects dose uniformity [139].

#### 5.1.4. Considerations in DPI Selection

DPIs selection among these different categories depends on several requirements, such as the dose, its frequency, and on the properties of the powder formulation. Reusable, reloadable, multi-dose inhalers are preferred when the therapy requires a frequent use of the device, while single-use devices are favored in innovative DPI developments such as antimicrobials, vaccines, and similar low dose frequency or single dose APIs [186]. The production of high-quality aerosols is significantly important and requires attention to various concepts extending to ergonomic factors as well to ensure reproducible dosing and generation of adequately inhalable formulations. Regardless of design improvements (such as the inclusion of spacers), improper use of pMDIs is still a common issue. DPIs do not require much, if any, coordination between inhalation and actuation because they are activated by the patient’s inspiratory airflow. Better lung delivery has frequently been obtained as a result than with comparable pMDIs. DPIs are generally composed of solid-particle blends in a single-phase, which makes them more desirable in terms of stability and processing. Due to their lower energy state and lower rate of chemical degradation, dry powders are less likely to react with contact surfaces. pMDI formulations, on the other hand, which include propellant and co-solvents, may extract organic compounds from device components [55]. The use of DPIs has greatly decreased the usage of nebulizers and MDIs. However, it is noteworthy that in the UK, where DPIs were invented, they only accounted for less than 30% of total respiratory retail units in 2017, while MDIs made up 70%. In almost every other European country, DPIs have a higher market share than MDIs, with Sweden having the highest DPI share at 85% compared to only 13% for MDIs. The popularity of MDIs and DPIs has significantly reduced the use of nebulizers in Europe, except in Italy, where they still account for 44% of retail sales. Although DPIs have advantages over MDIs and nebulizers, it is regrettable that device and formulation studies are still being carried out separately instead of being integrated for optimal results [64].

## 6. Advancements in Inhaler Devices: Ideal Characteristics and Marketed Innovations

### 6.1. Ideal Characteristics of Inhaler Devices

As seen previously, DPIs’ formulations have to answer to particular demands [190]. For instance, a comprehensive study conducted by Kolewe et al. investigated aerosol deposition in the upper airways of pediatric subjects, ranging from infants to 6-year-old children. The study focused on understanding the relationship between various parameters, such as particle size, flow rate, and anatomical factors like the glottis-to-cricoid ring diameter ratio (GC-ratio), epiglottis angle, and sex, and their influence on aerosol deposition. The findings of the study revealed that these parameters play a significant role in determining aerosol deposition in pediatric airways, emphasizing the need for inhalation devices designed specifically for pediatric patients to suit the anatomical and physiological differences in children’s airways [191].

Moreover, their drug administration will work effectively and safely only when the device delivers particles fine enough to deposit on the desired sites of action in the respiratory system. Consequently, the powder formulation as well as the model and the design of the device used play a significant role in meeting those demands [35].

The most suitable, ideal inhaler must meet specific standards and various criteria, including ease of use, affordability, portability, and ensuring that the drug remains stable. Regardless of their designs, DPIs undoubtedly represent a significant advancement in inhalation therapy since they meet most of the criteria listed below [131]. They offer several advantages over other inhaler types, such as achieving a higher pulmonary deposition than pMDIs [145]. Due to the lack of propellants, one significant advantage of DPIs is their environment sustainability. They require little to no patient coordination therefore making them appropriate for patients of all ages and abilities. Additionally, DPIs generally have a better formulation stability, ensuring the effectiveness of the medication throughout its shelf life. However, DPIs have certain limitations, such as a varying deposition effectiveness depending on patient airflow, with lung deposition ranging from 9% to 78.7% in currently available inhalers. Poor powder de-agglomeration, patient characteristics, cohesive forces in the formulation, and device design may all contribute to this variability [28]. DPIs also face challenges with dose uniformity, which affect treatment consistency. Finally, DPIs development and manufacturing are frequently more complex and expensive compared to other inhaler types [21,145]. That being said, the ideal criteria for an inhaling device are as follows:

User-friendly—Inhaler devices should prioritize ease of use, especially for patients like children or the elderly. A simple treatment method provides reassurance and convenience, reducing stress during acute attacks and preventing complications. Considering patient comfort is crucial when designing inhaler devices.

Informing and tracking—Inhaler devices should include a feedback mechanism to ensure accurate medication administration. Prompt feedback prevents under-dosing and double-dosing, providing peace of mind to patients and helping them track their medication usage for effective management of their condition.

Control—Inhaler devices should have a visible dose indicator to help patients manage medication usage effectively. This enables patients to anticipate refills, avoid unexpected shortages, and adhere to their treatment plan by providing reminders for timely medication intake.

Unnoticeable and portable—Inhaler devices should prioritize patient comfort and convenience. This includes discreet design for public use, compact size, lightweight, and portability for easy carrying. Focusing on these aspects empowers patients and improves treatment outcomes.

Hygienic—Inhaler devices should be easy to maintain, with a fast and hygienic loading process. The mouthpiece should be easily cleanable, especially for frequent use in various environments.

Dose delivery—To ensure effective inhaler devices, key factors include accurate and uniform dose delivery (throughout the life on the inhaler) across different inspiration flow rates, maintaining drug stability, accommodating a wide range of doses and drugs, and optimizing particle size for deep lung drug delivery.

All in all, significant research efforts are being devoted to optimizing DPIs and improving their features in order to be as close as possible to a device that would meet all the ideal criteria.

### 6.2. Current Marketed Inhalers and Inhalation Therapy Innovations

The inhalation delivery has witnessed a tremendous progress and improvement in the recent years to ameliorate daily regimens, patients’ compliance (notably by reducing the dose frequency), but also drug availability and delivery. Manufacturing processes, drug delivery strategies, as well as devices improvements have been made in this regard. An example of that would be lipid-based or polymer-based carriers in the formulation, as well as biotech drugs [192], with Afrezza^®^ for instance, the only protein (insulin) DPI, used for type 1 and 2 diabetes. As mentioned before, DPIs offer an encouraging approach to deliver biologics such as proteins, nucleic acids, viruses (like phages), and cells (attenuated bacterial cells such as Bacille Calmette–Guérin—BCG for tuberculosis). Proteins have received the most attention among biologics for inhalation but many of the protein formulation and manufacturing procedures have been modified for other biologics such as nucleic acids and, more recently, phages. DPIs can serve as a potential platform for inhaled gene therapies and vaccines, offering advantages such as mucosal immunization and a non-invasive injection-free administration [185].

The inhalation therapy constantly develops and innovates in order to bypass the different challenges it faces while still providing a high therapeutic effectiveness, and a wide range of DPIs are currently available in the market (Figure 10). Currently, there are 40 different inhaler device types available on the market, with DPIs being the most commonly used method to treat ailments, which affect over 500 million people worldwide [193,194].

Moreover, in order to obtain a more comprehensive overview, Figure 11 provides a step-by-step illustration of the proper use of a DPI device.

Moreover, newer and more advanced inhalation devices are able to supply a drug dose in the range of micrograms and sometimes nanograms instead of milligrams (Table 4), hence achieving a greater drug deposition: >50% lung deposition has been observed as opposed to ≤20% lung deposition with older inhalers [23,195]. It is worth noting that there are some DPIs carrying high doses, such as Relenza^®^ (zanamivir, 5 mg), Tobi Podhaler^®^ (tobramycin, 28 mg), Bronchitol (40 mg, mannitol), and Osmohale^®^ (mannitol) [196].

Moreover, effective management of respiratory conditions such as asthma and COPD heavily relies on the appropriate use of the prescribed inhaler. An inaccurate administration technique or flawed inhalation routine can result in reduced drug delivery, subsequently impacting the control of the disease. As mentioned previously, several studies showed that a faulty coordination between the patient’s inhalation and the device’s often leads to patients mishandling these devices. Despite the availability of numerous precise inhalation devices in the market, delivering an accurate dose can still be a challenge at times. As a result, spacers, also known as holding chambers or extension devices, can be employed to enhance the performance of pMDIs. Spacers are supplementary devices that function by enclosing the medication in a confined space during inhalation. When combined with proper inhalation technique, using a spacer can be a great way to increase drug delivery. A study conducted in India involving 300 patients revealed that 82.3% of them made at least one mistake while using their inhaler. Among these patients, the highest number of errors was observed in those using MDIs (94.3%). However, the use of a spacer with MDIs led to a lower percentage of errors (78%). Errors made by DPIs users amounted to 82.3%, followed by nebulizers users with 70% [178]. Therefore, using a spacer not only reduces the deposition in the oropharyngeal region but also reduces the need for a precise coordination of actuation and inhalation required when using a pMDI device on its own [199]. This is especially beneficial for infants and children who may lack the ability to perform a precise inhalation maneuver or may refuse to cooperate, or patients who require medical assistance, such as elderly patients with COPD and cognitive impairment. DPIs on the other hand do not need a spacer, nor to be shaken before each use.

As the field of inhalation therapy continues to advance with the development of more sophisticated devices and strategies to enhance drug delivery, the integration of digital smart inhalers marks a significant leap in the evolution of respiratory care. These innovative technologies use connectivity and data analysis to offer a real-time monitoring of the inhaler’s usage, providing valuable insights such as patient adherence. By seamlessly integrating digital solutions into traditional inhalation therapies, healthcare providers can further optimize treatment outcomes and empower patients to manage their respiratory conditions effectively. Digital inhalers are particularly beneficial for individuals struggling with such respiratory diseases. They are breathing aids which can be linked to electronic devices to check on a patient’s health every day. These are especially helpful for people with respiratory diseases like asthma or obstructive diseases. Sensors in these inhalers prompt the user to take medicine at the appropriate times each day. Digital inhaler health systems use electromechanical sensors and microelectronics to monitor inhaler actuation. They are marketed as the first and only line of smart inhalers with integrated sensors that automatically log the patient’s inhalations: every time the patient opens the cap or takes a breath, it is recorded as an inhaler use event. The inhalation pattern is then saved and can be used to assist patient and doctor in personalizing the treatment plan [200].

Another great innovation is the Propeller Health sensor. This sensor seamlessly attaches to conventional inhalers, transforming them into smart inhalers by passively tracking and recording medication usage. The sensor data is then sent to the Propeller mobile app or online portal (using Bluetooth or cellular connectivity) for the purposes of collection, analyzing, and/or sharing [201]. Adherium’s Hailie^®^ is another example of a cloud-based platform that gathers inhaler use data from Bluetooth sensors equipped onto standard inhalers, providing real-time feedback, helping monitor and improve adherence through their mobile app, and even providing missed-doses reminders [202].

A study by Van Sickle et al. confirmed the efficacy of digital smart inhalers in revolutionizing respiratory care by providing real-time monitoring and insights into patient adherence. Using the mobile health program resulted in improvements in asthma outcomes including better adherence to medications, more asthma-free days, and improved overall asthma control [203]. Patient involvement and engagement will undoubtedly rise with the use of smart DPIs and associated apps. If implemented effectively, it will support patients in their self-management and contribute to a healthcare system that is more patient-centered. The key challenges that still need to be overcome are the complex business models and the lack of interoperability standardization [204].

Another interesting innovation to note is dry powder inhalation being explored for nose-to-brain targeting, aiming to enhance drug delivery to the central nervous system (CNS). Notably, the investigation of TS-002 dry powder demonstrated significantly improved bioavailability in cynomolgus monkeys. Similar positive outcomes were observed in separate studies involving donepezil and dexamethasone. These findings underscore the potential of DPIs in optimizing drug delivery to the CNS, opening new avenues for effective therapeutic applications [193].

### 6.3. Biocompatibility of DPIs

The field of inhalation therapy, while focusing on the efficacy and safety of inhalers in managing respiratory diseases, must also give due attention to biocompatibility. Biocompatibility refers to the compatibility of a medical device with the biological system without causing harm. In the context of inhalers, ensuring biocompatibility is crucial to avoid adverse reactions or health risks associated with the materials used in the device. The biocompatibility of DPIs is an important consideration in their design and development. According to the International Organization for Standardization (ISO), ISO 18562-1:2017 specifies general criteria for assessing the biocompatibility of medical device materials. While ISO 18562 testing is a requirement for MDIs, it is not for DPIs [205]. However, in vitro cytotoxicity tests have been conducted to evaluate the biocompatibility of DPI medical devices. This testing is particularly significant due to the prolonged contact of the DPI mouthpieces with users on a daily basis. In a comprehensive study, the cytotoxicity of the mouthpieces of four marketed DPIs: Aerolizer^®^ (Novartis), Diskus^®^ (GlaxoSmithKline), Elpenhaler^®^ (Elpen Pharma), and Turbuhaler^®^ (AstraZeneca) was evaluated. The experimental procedure adhered to guidelines from the ISO, the FDA, and the United States Pharmacopeia (USP). The results indicated that all DPI mouthpieces were found to be equivalently safe for long-term use. The methods presented in the study offer an easy and sensitive way for the test of cytotoxicity of a biomaterial and can be applied to other medical devices [206].

Furthermore, considerations extend beyond the device itself to the formulation of DPIs. Natural and bioinspired excipients for DPI formulations were also discussed by Zillen et al. The article explores various excipients in dry powder inhalers (DPIs) for lung medication, emphasizing their safety. It covers amino acids, sugars, lipids, and biodegradable polymers. The authors stress the challenge of the body clearing these substances from the lungs and highlight the need for more research on their potential toxicity. They emphasize that information regarding the pulmonary toxicity of excipients in DPI formulations is generally lacking [207].

As innovation goes, the use of hyaluronic acid (HA) represents a cutting-edge approach, as it suggests a conscious effort to leverage a naturally occurring substance with potential benefits for the lungs. PolmonYDEFENCE/DYFESA™ is a novel formulation based on HA delivered to the airways using the PillHaler^®^ DPI device. The unique feature involves HA molecules creating a protective barrier on cell surfaces. Importantly, in animal models, exposure to the device has been demonstrated to be safe [208].

Although toxicity studies may be costly, expanding our understanding of excipients’ toxicological aspects holds the potential to significantly accelerate progress in pulmonary therapeutics. A more in-depth comprehension of excipients, their mechanisms, and optimal combinations is crucial to advance in this field [207]. As inhaler technologies continue to evolve, integrating biocompatibility assessments into the design process becomes imperative. Manufacturers and researchers must collaborate to enhance not only the therapeutic efficacy of inhalers but also their safety profile through comprehensive biocompatibility studies.

### 6.4. Significance of the Evolution of Inhaler Technology

The evolution of inhaler technology through new powder processing methods and inhaler devices marks a significant step in enhancing inhalation therapies. These advancements bring forth improvements in inhaler efficiency, dispersion, patient adherence, and provided alternative dosage forms. However, it is crucial to recognize that inhaler devices, while offering notable benefits, may not be as straightforward in their use as commonly perceived by both clinicians and patients. This realization emphasizes the importance of an understanding of inhaler ease of use, a factor linked to patient adherence and, consequently, treatment efficacy [209].

#### 6.4.1. Improved Efficiency

Innovations in powder processing methods and inhaler devices play a crucial role in advancing the efficiency of aerosol drug delivery. The enhanced effectiveness of these modern devices allows for comparable therapeutic benefits with reduced nominal drug doses. This progress is driven by advancements in technology that aim to optimize inhaler performance through factors such as particle size, formulation, and device design. Specifically, modern DPIs with innovative designs are capable of generating fine powder aerosols, thereby significantly enhancing drug delivery to the lungs. Furthermore, the evolution of active or power-assisted DPIs using sophisticated mechanisms contributes to efficient drug dispersion (thus efficiency), decreasing the dependency on high inspiratory flow rates from patients. As these technologies progress, electronic DPIs, for instance the MicroDose^®^, demonstrate the potential to incorporate features like dose delivery confirmation, adherence monitoring, and dosing reminders, marking a promising future for inhaler technology [209].

#### 6.4.2. Improved Dispersion

Recent advancements in DPIs have propelled the evolution of dispersion mechanisms to enhance inhaler performance. Some newly developed DPIs, along with existing devices used for new powder formulations, continue to employ low-resistance capsule-based systems. However, these encounter challenges in optimizing powder properties for both capsule emptying and effective dispersion. However, it is worth noting that certain inhalers strategically employ advanced dispersion systems to ensure the efficient de-agglomeration of the inhaled powder. Some modern DPIs use innovative technologies such as vibrating piezo-electric crystals, battery-driven impellers, and electronic components. These enhancements ensure not only efficient de-agglomeration of the inhaled formulation but also provide increased precision in dosing and reproducible aerosol production [209].

#### 6.4.3. Improved Patient Adherence

Ensuring consistent adherence to prescribed inhaler medication and using accurate inhaler techniques play crucial roles in determining the efficacy of asthma and COPD treatments. It is important to emphasize the critical impact of regular and proper inhaler use on asthma outcomes, where adherence to prescribed treatment plans leads to favorable results, including reduced symptoms, fewer exacerbations, and improved quality of life [210]. Additionally, the integration of modern, new digital smart inhalers represents a significant step forward. As previously discussed, the validated effectiveness of digital smart inhalers improved respiratory treatments [203]. Digital inhalers, equipped with sensors, not only prompt users for timely medication but also automatically log each inhalation event, resulting in improved asthma outcomes and overall disease control.

#### 6.4.4. Alternative Dosage Forms

Examining the past history of DPIs reveals that their early conceptualization predominantly focused on delivering low drug doses for asthma and COPD [8]. Furthermore, high-dose DPIs emerge as a viable alternative dosage form to conventional nebulized medications. The progression in dry powder formulations, coupled with the evolution of advanced DPI variants capable of delivering doses exceeding 100 mg, indicates a significant evolution in dosage forms. Moreover, the discussion on combination therapy for respiratory conditions such as asthma and COPD illustrates the versatility of these inhalation systems, offering diverse and effective options for patients. These findings contribute to a broader understanding of how emerging powder processing methods and DPIs impact inhalation drug delivery [211]. DPIs remain a significant and actively researched method for addressing a widening range of respiratory issues. Furthermore, the exploration of various DPIs, along with new advancements in powder processing methods, demonstrates their importance in modern therapeutic strategies [212].

## 7. Conclusions

With the continuous advancement in DPI technology, the field of inhalation therapy is witnessing a remarkable rise in clinical research to explore its potential further. Numerous ongoing clinical trials are actively investigating the efficacy and safety of DPIs in managing respiratory diseases. Presently, four studies are actively recruiting participants, while an additional 15 trials have already commenced recruitment [10]. Moreover, as macromolecules and biotech medicines become increasingly relevant, their inclusion promises to transform the field of DPIs, providing innovative approaches and targeted, increased therapy options for patients with respiratory disorders. These continuous efforts highlight the growing interest in DPI research, paving the way for possible future breakthroughs and improvements in respiratory healthcare.

## Figures and Tables

**Figure 1 pharmaceuticals-16-01658-f001:**
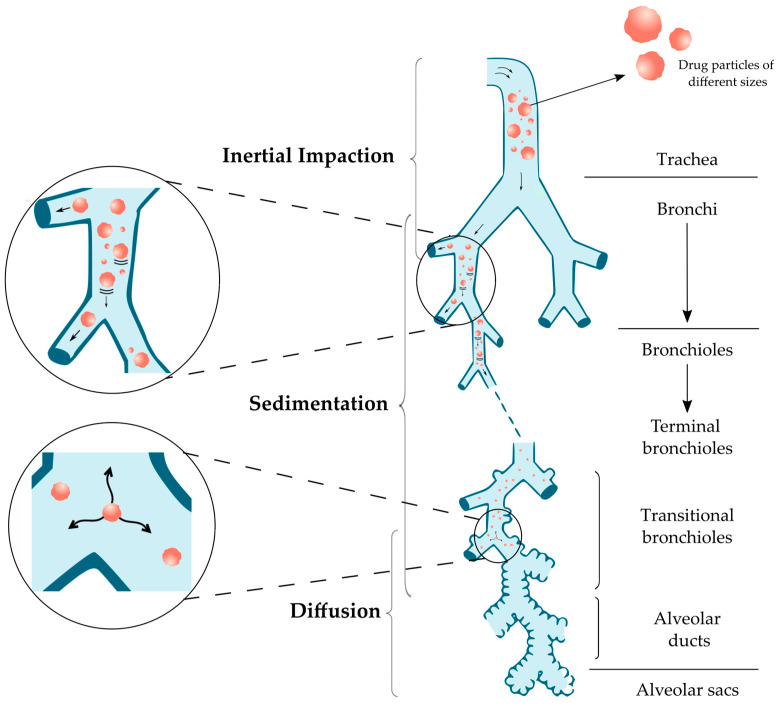
Particle deposition mechanisms according to particle size.

**Figure 2 pharmaceuticals-16-01658-f002:**
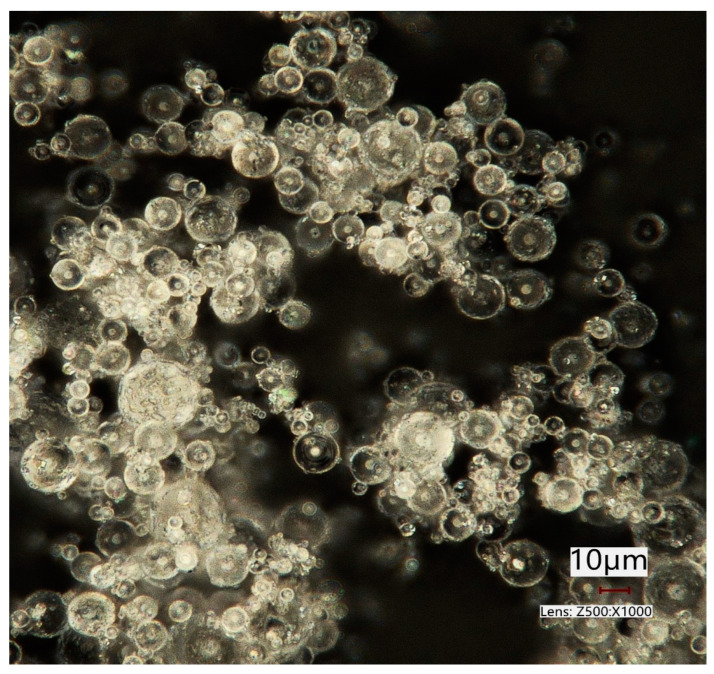
Spray dried lactose particles. Image taken with a Keyence VHX 970F digital microscope (Keyence Corp., Osaka, Japan).

**Figure 3 pharmaceuticals-16-01658-f003:**
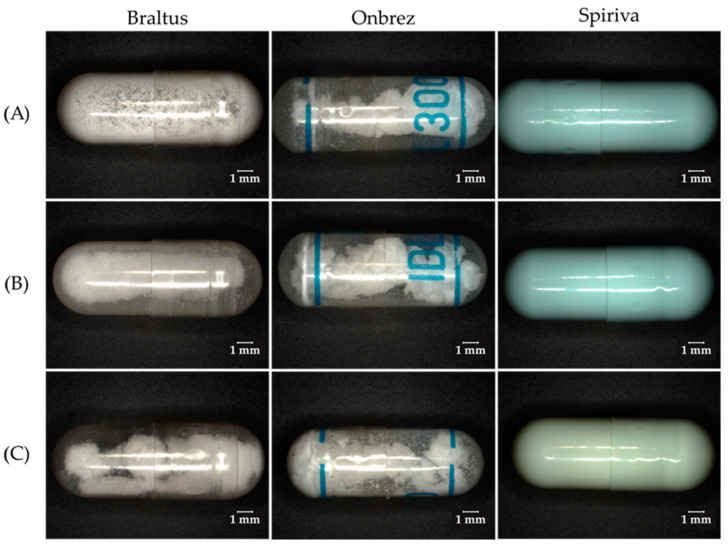
Visual assessment of DPI (Braltus^®^, Onbrez^®^, Spiriva^®^) capsules stability under varying environmental conditions: from the packaging (**A**), after 7 days at room temperature (**B**), and after 7 days in set conditions (**C**).

**Figure 4 pharmaceuticals-16-01658-f004:**
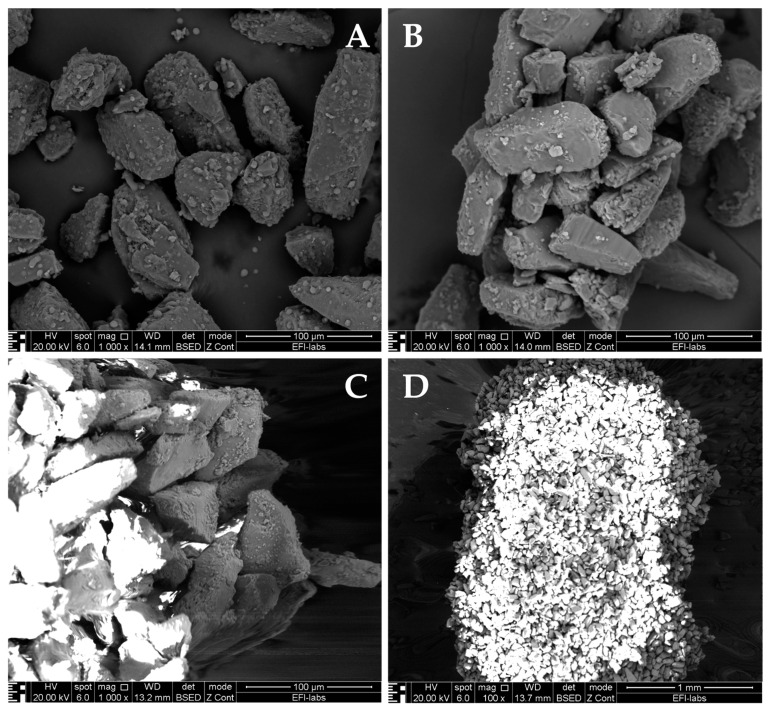
Scanning Electron Microscopic images of Braltus^®^ capsules powder formulations from their packaging (**A**) and after 3 (**B**) and 7 (**C**,**D**) days in set conditions. Magnification: (**A**–**C**): 1000×; (**D**): 100×.

**Figure 5 pharmaceuticals-16-01658-f005:**
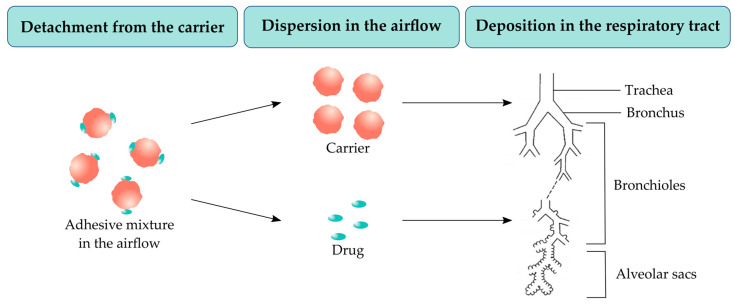
Drug delivery steps of adhesive mixtures after inhalation (based on [151]).

**Figure 6 pharmaceuticals-16-01658-f006:**
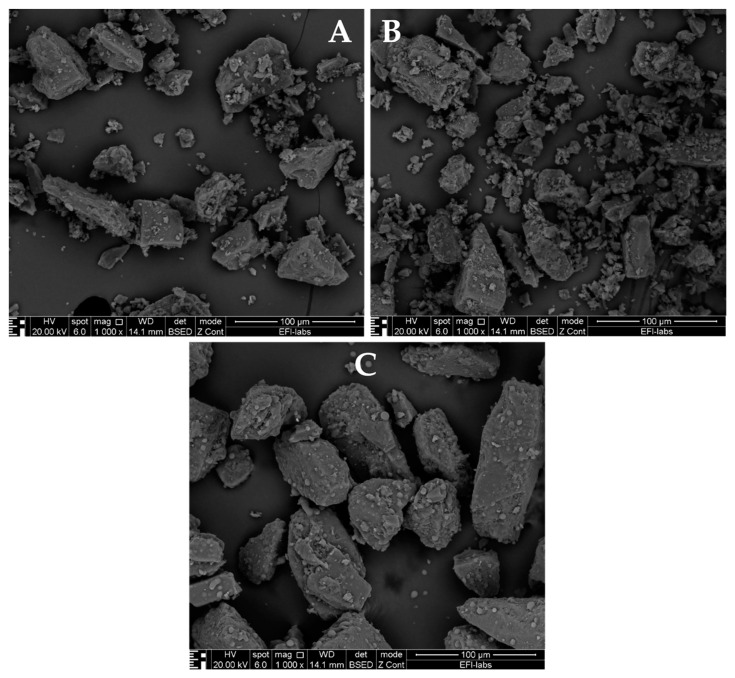
SEM pictures of DPI capsules powder formulations directly from their packaging: Spiriva^®^ (**A**), Onbrez^®^ (**B**), and Braltus^®^ (**C**). Magnification: 1000×.

**Figure 7 pharmaceuticals-16-01658-f007:**
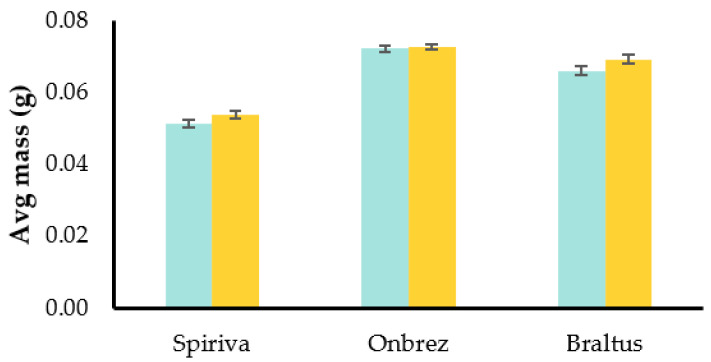
Average mass of DPI capsules straight from their packaging (day 0, in cyan) and after being left at room temperature for 7 days (in yellow).

**Figure 8 pharmaceuticals-16-01658-f008:**
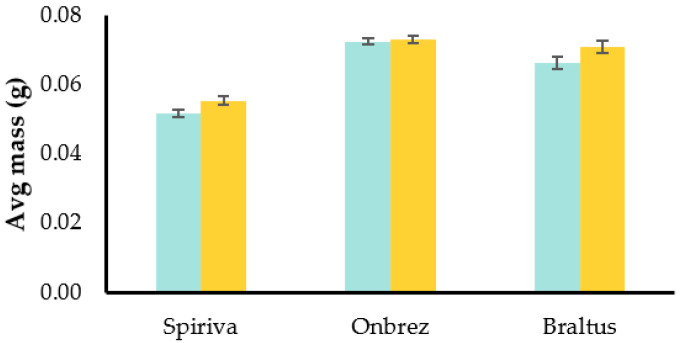
Average mass of DPI capsules straight from their packaging (day 0, in cyan) and after being left in a stability chamber for 7 days (in yellow).

**Figure 9 pharmaceuticals-16-01658-f009:**
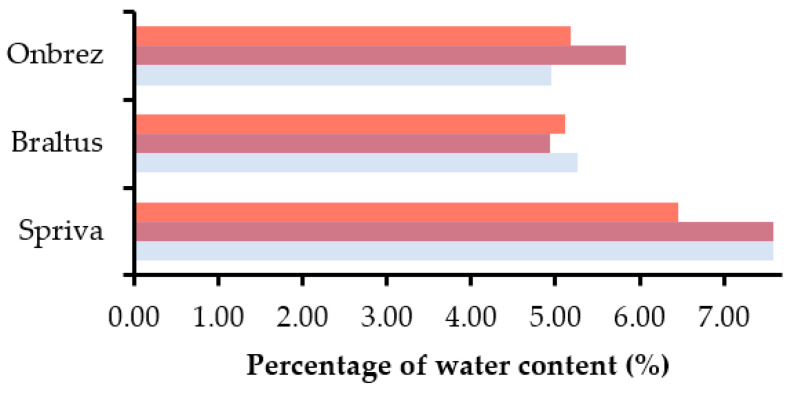
Percentages of DPI powder formulations water content straight using Karl Fischer titration directly out of their packaging (in gray), after 7 days at room temperature (purple), and after 7 days in a stability chamber with set conditions (pink).

**Figure 10 pharmaceuticals-16-01658-f010:**
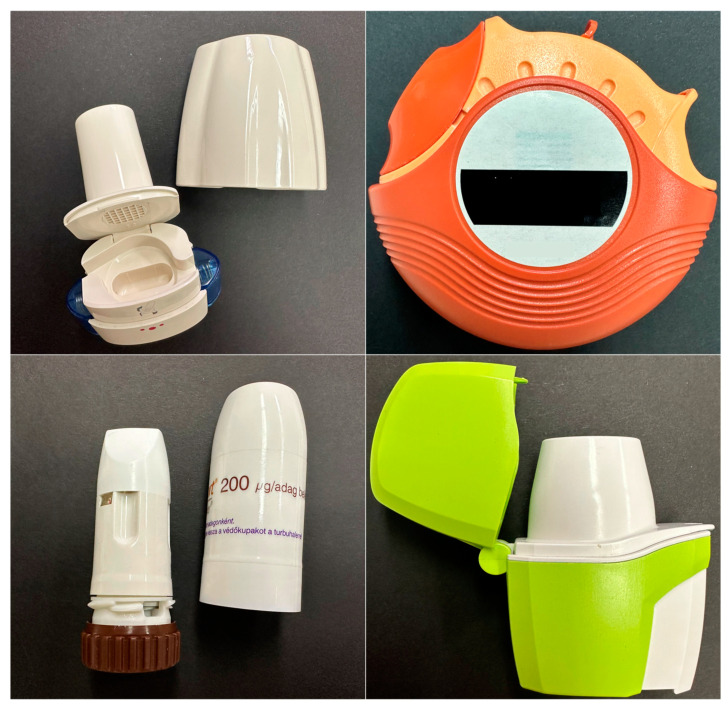
Examples of currently marketed DPI devices.

**Figure 11 pharmaceuticals-16-01658-f011:**
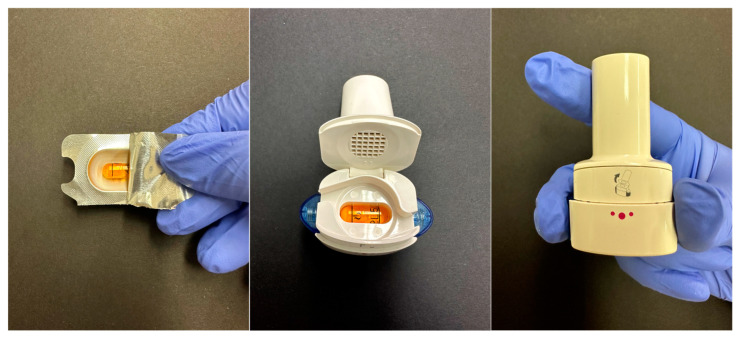
Step-by-step demonstration of the use of a Breezhaler^®^ DPI device: remove the capsule from the blister, place it in the device, close it and push the side buttons to puncture the capsule to release the powder.

**Table 1 pharmaceuticals-16-01658-t001:** Deposition mechanisms in the lungs according to particle size.

Particle Size	Mechanism	Parts of Respiratory Tract
Above 5 μm	Inertial impaction	Oropharynx and conducting airways
0.5–5 μm	Sedimentation	Bronchi, Bronchioles and Alveoli
0.5–3 μm	Sedimentation and Diffusion	
Below 0.5 μm	Diffusion and Brownian motion	Alveolar region

**Table 2 pharmaceuticals-16-01658-t002:** Inhalable particles in dry powder formulations.

Type of Particles	Characteristic	Active Pharmaceutical Ingredient	Method of Preparation	Size	Ref.
Polymeric microparticles (MPs)	Chitosan MPs	Rifabutin and Rifampicin	Spray drying	1–5 μm	[70]
	Locust bean gum (LBG) MPs	Isoniazid or Rifabutin		1.15–1.67 μm	[71]
	PLGA ^1^ MPs	Recombinant human interleukin-2 (rhIL-2)	Modified w/o/w double emulsion solvent extraction method	4.02 μm	[72]
		Heparin	Spray drying	2.55–3.86 µm	[73]
		Rifapentine		~2 µm	[74]
		Bovine serum albumin (BSA) as a model vaccine	Supercritical CO_2_-assisted spray-drying (SASD)	1.7–3.5 µm	[75]
		Rifampicin-loaded microspheres	Solvent evaporation method with premix membrane homogenization	0.64–4.1 µm	[76]
	Modified PLGA	siRNA	Double emulsion solvent evaporation method	207.7–261.1 nm	[77]
	PCL ^2^ MPs	Resveratrol	Vibrational atomization spray drying	3.8 µm	[78]
Microparticles	Solid Lipid Microparticles (SLMs)	Quercetin	o/w emulsification method	5.72 µm	[79]
				2.90 µm	[80]
	None	Naringin	Spray drying	3.29–3.92 µm	[81]
	Sprayed with amino acids		Spray drying	2.75–3.42 µm	[82]
	None	Atropine	Solid-phase extraction	3.7 µm	[83]
Porous particles	Large porous particles (LPPs)	Doxorubicin	w/o/w double emulsion method	14.1 µm	[84]
		Celecoxib	PLGA LPPs by supercritical fluid pressure-quench technology	10.53 µm	[85]
	PLGA-based gas-foamed LPPs	Rhodamine B isothiocyanate–dextran	Double emulsion solvent evaporation method	~30 μm	[86]
Porous particles	Nanocrystals embedded in microparticles	Niclosamide	Spray freeze drying	0.18–4.29 µm	[87]
Swellable particles	Hydrogel microparticles	Paclitaxel	Emulsification/gelation method	<5 µm	[88]
		BSA as a model protein	Spray drying	3.6 µm	[89]
Swellable particles	Hydrogel microparticles	Ciprofloxacin and Doxycycline	Spray drying	~2 μm	[90]
		Chemotherapeutic drugs		~6 μm	[91]
		Matrix metalloproteinase (MMP) enzyme-responsive hydrogel	Modified polymerization method	2.8–4.0 μm	[92]
Nanoparticles	Proliposomes	Rifapentine	Spray drying	7.73 μm	[93]
	Liposomes	Synergistic Ciprofloxacin and Colistin	Ultrasonic sprayfreeze drying	~100 μm	[94]
		Ciprofloxacin	Membrane extrusion of multilamellar liposomes followed by remote loading of API	3.6–4.0 µm	[95]
		Isoniazid	Thin-film hydration method	755 nm	[96]
		Insulin		100 nm	[97]
		Oseltamivir phosphate	Spray drying	3.5 µm	[98]
		Curcumin	Nano-spray drying	2.10 µm	[99]
		Gemcitabine hydrochloride	Lyophilisation	325 nm	[100]
		Salbutamol sulfate	Vesicular phospholipid gel (VPG) technique	~10 μm	[101]
				57 nm	[102]
		Rifampicin	Chloroform-film method, lyophilisation	200–300 nm	[103]
Nanoparticles	Liposomes	Tacrolimus	Thin film evaporation, spray drying	9.46–12.4 μm	[104]
		Dapsone		7.9–11.2 μm	[105]
		Curcumin	Film method	94.65 nm	[106]
	Solid lipid nanoparticles (SLNs) ^3^	Alendronate	Homogenization	<100 nm	[107]
		Amikacin		164 nm	[108]
	Solid lipid nanoparticles (SLNs) ^3^	Doxorubicin	Homogenization	94–113 nm (with d triethanolamine) 127–151 nm (with stearylamine)	[109]
		Insulin	w/o/w emulsion	231.67 nm	[110]
	Solid lipid nanoparticles (SLNs) ^3^	Rifampicin	Melt emulsifying technique then freeze drying	0.47–1.72 mm	[111]
		Budesonide	Emulsification-solvent diffusion method	218.2 nm	[112]
	Polymeric nanoparticles	Sildenafil	Vibrational spray drying	~4–8 μm	[113]
		N-acetylcysteine	w/o/w double emulsion	307.50 nm	[114]
		Heparin	Ionotropic gelation technique	162–217 nm	[115]
		Fisetin	Spray drying	1.5 µm	[116]
	Protein-based nanoparticles	Apigenin	Spray drying	376 nm	[117]
	Nanocomposite particles	Curcumin	Spray drying	2.1 µm	[118]
		Andrographolide		3.37 µm	[119]
		Salvianolic acids	Freeze drying	<5 µm	[120]
	Porous nanoparticle-aggregate particles	Rifampicin	Spray drying	195 nm	[121]
		Levofloxacin	Spray freeze drying	18 µm	[122]
	Nanostructured lipid carrier (NLC)	Montelukast	Lyophilization	184.6 nm	[123]
		Paclitaxel	Emulsification and ultrasonication method, spray drying	283.4 nm	[124]
Nanoparticles	Nanoparticle agglomerates	Nifedipine	Solvent precipitation, controlled particle agglomeration, lyophilization	470 nm	[125]
	Nanocrystals	Curcumin	Spray drying	924 nm	[126]
			Supercritical (ARISE) processing	3–5 µm	[127]
		Baicalein	Modified anti-solvent recrystallization then high pressure homogenization	Not specified	[128]
Microspheres	Technosphere^®^	Insulin	Precipitation, micoencapsulation	2–5 µm	[129]
	PulmoSphere™	Tobramycin	Emulsion-based spray drying	1–5 μm	[130]
	iSPERESE™	Tiotropium bromide	iSPERSE dry powder delivery technology	~3 μm ^4^	[131]
	Polyamidoamine (PAMPAM) dendrimers	Rifampicin	Spray drying	~6 μm	[132]
	Spherical particles	Curcumin	Spray drying	1–5 μm	[133]
Dendrimers	siRNA–dendrimer nanocomplexes	siRNA	Microfluidics, Spray-drying	Not specified	[134]
	Doxorubicin–PAMAM dendrimer conjugate loaded with mannitol microparticles	Doxorubicin	Spray drying	1 µm	[135]

^1^ PLGA: poly(lactic-co-glycolic acid).^2^ PCL: poly(ε-caprolactone).^3^ The small particle size of SLNs (<100 nm) makes them unsuitable for use as standalone DPIs. To address this limitation, researchers have explored methods to incorporate SLNs into larger carriers or mix them with inert bulking agents like mannitol, dextran, or lactose. This process aims to reach an aerodynamic size range of 1–5 μm, which is suitable for DPIs. One commonly utilized technique for achieving this modification is the spray drying method [136].^4^ Aerodynamic particle size.

**Table 3 pharmaceuticals-16-01658-t003:** DPI formulations common testing (based on [16,159,160]).

Test	Description
Particle size determination	The determination is executed using a cascade impactor or using a light scattering decay method. The particle size is expressed in µm.
InVitro Aerodynamic Assessment	Evaluates the aerodynamic behavior of emitted particles, considering factors like the Mass Median Aerodynamic Diameter (MMAD).
Fine Particle Fraction (FPF)	Measures the fraction of fine particles (usually below 5 µm) that are emitted from the DPI, indicating their suitability for deep lung deposition.
Delivered dose	Ensures the delivered dose per actuation matches the intended dose and meets regulatory requirements.
Dose uniformity	Ensures uniformity of the dose by weighing the container before and after a specific number of actuations. The difference in weight per dose is calculated.
Content uniformity	Assesses the uniform distribution of the active pharmaceutical ingredient (API) within the DPI formulation.
Moisture content	Measures the moisture content using methods such as Karl-Fischer or gas chromatography.
Bulk density	Determines the bulk density of the DPI formulation using methods like pycnometry.
Tapped density	Measures the tapped density, which assesses the powder’s ability to pack and flow effectively.
Flowability	Evaluates the flow properties of the DPI formulation, which can affect device metering and aerosol dispersion.

**Table 4 pharmaceuticals-16-01658-t004:** Marketed inhalers listed according to their delivered dose (the dose leaving the mouthpiece of the inhaler) [196,197,198].

Inhaler/Medicine Name	API ^1^ Dose Quantity and Name	Carrier Excipient	Company
Oxis Turbohaler 6	4.5 µg Formoterol fumarate dihydrate	Lactose monohydrate	AstraZeneca UK Ltd., London, UK.
Oxis Turbohaler 12	9 µg Formoterol fumarate dihydrate	Lactose monohydrate	AstraZeneca UK Ltd., London, UK.
Braltus 10 (with Zonda inhaler device)	10 µg Tiotropium	Lactose monohydrate	Teva UK Ltd., Harlow, UK.
Spiriva 18 (with HandiHaler device)	10 µg Tiotropium	Lactose monohydrate	Boehringer Ingelheim Ltd., Ingelheim, Germany.
Tiogiva 18	10 µg Tiotropium	Lactose monohydrate	Glenmark Pharmaceuticals Ltd., Mumbai, India.
Foradil 12	12 µg Formoterol fumarate dihydrate	Lactose monohydrate	Novartis Pharmaceuticals UK Ltd., London, UK.
Formoterol Easyhaler 12	12 µg Formoterol fumarate dihydrate	Lactose monohydrate	Orion Pharma UK Ltd., Reading, UK.
Acopair 18 (with NeumoHaler device)	12 µg Tiotropium	Lactose anhydrous	Mylan, Hatfield, UK.
Flixotide Accuhaler 50	50 µg Fluticasone propionate	Lactose monohydrate	GlaxoSmithKline UK Ltd., London, UK.
Serevent Accuhaler 50	50 µg Salmeterol xinafoate	Lactose monohydrate	GlaxoSmithKline UK Ltd., London, UK.
Seebri Breezhaler 44	55 µg Glycopyrronium bromide	Lactose monohydrate	Novartis Pharmaceuticals UK Ltd., London, UK.
Incruse Ellipta 55	55 µg Umeclidinium	Lactose monohydrate	GlaxoSmithKline UK Ltd., London, UK.
Fobumix Easyhaler 80/4.5	84.5 µg Budesonide (80 µg) Formoterol fumarate dihydrate (4.5 µg)	Lactose monohydrate	Orion Pharma UK Ltd., Reading, UK.
Fostair NEXThaler 100/6	86.9 µg Beclometasone dipropionate anhydrous (81.9 µg) Formoterol fumarate dihydrate (5 µg)	Lactose monohydrate	Chiesi Ltd., Parma, Italy.
Symbicort Turbohaler 100/6	84.5 µg Budesonide (80 µg) Formoterol fumarate dihydrate (4.5 µg)	Lactose monohydrate	AstraZeneca UK Ltd., London, UK.
Anoro Ellipta 55/22	87 µg Umeclidinium bromide (65 µg) Vilanterol trifenatate (22 µg)	Lactose monohydrate	GlaxoSmithKline UK Ltd., London, UK.
Flixotide Accuhaler 100	100 µg Fluticasone propionate	Lactose monohydrate	GlaxoSmithKline UK Ltd., London, UK.
Pulmicort Turbohaler 100	100 µg Budesonide	None	AstraZeneca UK Ltd., London, UK.
Easyhaler Budesonide 100	100 µg Budesonide	Lactose monohydrate	Orion Pharma UK Ltd., Reading, UK.
Easyhaler Salbutamol 100	100 µg Salbutamol sulfate	Lactose monohydrate	Orion Pharma UK Ltd., Reading, UK.
Salbulin Novolizer 100	100 µg Salbutamol sulfate	Lactose monohydrate	Mylan, Hatfield, UK.
Trimbow NEXThaler 88/5/9	102 µg Beclometasone dipropionate (88 µg) Formoterol fumarate dihydrate (5 µg) Glycopyrronium (9 µg)	Lactose monohydrate	Chiesi Ltd., Parma, Italy.
Seffalair Spiromax 100/12.75	112.75 µg Fluticasone propionate (100 µg) Salmeterol xinafoate (12.75 µg)	Lactose monohydrate	Teva UK Ltd., Harlow, UK.
Relvar Ellipta 92/22	114 µg Fluticasone furoate (92 µg) Vilanterol trifenatate (22 µg)	Lactose monohydrate	GlaxoSmithKline UK Ltd., London, UK.
Onbrez Breezhaler 150	120 µg Indacaterol maleate	Lactose monohydrate	Novartis Pharmaceuticals UK Ltd., London, UK.
Seretide 100 Accuhaler	139 µg Fluticasone propionate (92 µg) Salmeterol xinafoate (47 µg)	Lactose monohydrate	GlaxoSmithKline UK Ltd., London, UK.
Fixkoh Airmaster 50/100	139 µg Fluticasone propionate (92 µg) Salmeterol xinafoate (47 µg)	Lactose monohydrate	Genus Pharmaceuticals, Huddersfield, UK.
Ultibro Breezhaler 85/43	164 µg Indacaterol maleate (110 µg) Glycopyrronium bromide (54 µg)	Lactose monohydrate	Novartis Pharmaceuticals UK Ltd., London, UK.
Fobumix Easyhaler 160/4.5	164.5 µg Budesonide (160µg) Formoterol fumarate dihydrate (4.5 µg)	Lactose monohydrate	Orion Pharma UK Ltd., Reading, UK.
WockAir 160/4.5	164.5 µg Budesonide (160 µg) Formoterol fumarate dihydrate (4.5 µg)	Lactose monohydrate	Wockhardt UK Ltd., Wrexham, UK.
Symbicort Turbohaler 200/6	164.5 µg Budesonide (160 µg) Formoterol fumarate dihydrate (4.5 µg)	Lactose monohydrate	AstraZeneca UK Ltd., London, UK.
DuoResp Spiromax 160/4.5	164.5 µg Budesonide (160 µg) Formoterol fumarate dihydrate (4.5 µg)	Lactose monohydrate	Teva UK Ltd., Harlow, UK.
Fostair NEXThaler 200/6	164.8 µg Beclometasone dipropionate anhydrous (158.8 µg) Formoterol fumarate dihydrate (6 µg)	Lactose monohydrate	Chiesi Ltd., Parma, Italy.
Trelegy Ellipta 92/55/22	179 µg Fluticasone furoate (92 µg) Umeclidinium bromide (65 µg) Vilanterol trifenatate (22 µg)	Lactose monohydrate	GlaxoSmithKline UK Ltd., London, UK.
Easyhaler Beclometasone 200	180 µg Beclometasone dipropionate	Lactose monohydrate	Orion Pharma UK Ltd., Reading, UK.
Atectura Breezhaler 125/62.5	187.5 µg Indacaterol acetate (125 µg) Mometasone furoate (62.5 µg)	Lactose monohydrate	Novartis Pharmaceuticals UK Ltd., London, UK.
Pulmicort Turbohaler 200	200 µg Budesonide	None	AstraZeneca UK Ltd., London, UK.
Easyhaler Budesonide 200	200 µg Budesonide	Lactose monohydrate	Orion Pharma UK Ltd., Reading, UK.
Easyhaler Salbutamol 200	200 µg Salbutamol sulfate	Lactose monohydrate	Orion Pharma UK Ltd., Reading, UK.
Budelin Novolizer	200 µg Budesonide	Lactose monohydrate	Mylan, Hatfield, UK.
Asmanex Twisthaler 200	200 µg Mometasone furoate	Lactose anhydrous	Organon Pharma UK Ltd., London, UK.
Ventolin Accuhaler 200	200 µg Salbutamol sulfate	Lactose monohydrate	GlaxoSmithKline UK Ltd., London, UK.
Relvar Ellipta 184/22	206 µg Fluticasone furoate (184 µg) Vilanterol trifenatate (22 µg)	Lactose monohydrate	GlaxoSmithKline UK Ltd., London, UK.
Seffalair Spiromax 202/12.75	214.75 µg Fluticasone propionate (202 µg) Salmeterol xinafoate (12.75 µg)	Lactose monohydrate	Teva UK Ltd., Harlow, UK.
Onbrez Breezhaler 300	240 µg Indacaterol maleate	Lactose monohydrate	Novartis Pharmaceuticals UK Ltd., London, UK.
Flixotide Accuhaler 250	250 µg Fluticasone propionate	Lactose monohydrate	GlaxoSmithKline UK Ltd., London, UK.
Atectura Breezhaler 125/127.5	252.5 µg Indacaterol acetate (125 µg) Mometasone furoate (127.5 µg)	Lactose monohydrate	Novartis Pharmaceuticals UK Ltd., London, UK.
Fixkoh Airmaster 50/250	274 µg Fluticasone propionate (229 µg) Salmeterol xinafoate (45 µg)	Lactose monohydrate	Genus Pharmaceuticals, Huddersfield, UK.
Seretide 250 Accuhaler	278 µg Fluticasone propionate (231 µg) Salmeterol xinafoate (47 µg)	Lactose monohydrate	GlaxoSmithKline UK Ltd., London, UK.
Sereflo Ciphaler 50/250	278 µg Fluticasone propionate (231 µg) Salmeterol xinafoate (47 µg)	Lactose monohydrate	Cipla EU Ltd., Addlestone, UK.
Fusacomb Easyhaler 50/250	286 µg Fluticasone propionate (238 µg) Salmeterol xinafoate (48 µg)	Lactose monohydrate	Orion Pharma UK Ltd., Reading, UK.
Enerzair Breezhaler	308 µg Indacaterol acetate (114 µg) Glycopyrronium bromide (58 µg) Mometasone furoate (136 µg)	Lactose monohydrate	Novartis Pharmaceuticals UK Ltd., London, UK.
Fobumix Easyhaler 320/9	329 µg Budesonide (320 µg) Formoterol fumarate dihydrate (9 µg)	Lactose monohydrate	Orion Pharma UK Ltd., Reading, UK.
WockAir 320/9	329 µg Budesonide (320 µg) Formoterol fumarate dihydrate (9 µg)	Lactose monohydrate	Wockhardt UK Ltd., Wrexham, UK.
Symbicort Turbohaler 400/12	329 µg Budesonide (320 µg) Formoterol fumarate dihydrate (9 µg)	Lactose monohydrate	AstraZeneca UK Ltd., London, UK.
DuoResp Spiromax 320/9	329 µg Budesonide (320 µg) Formoterol fumarate dihydrate (9 µg)	Lactose monohydrate	Teva UK Ltd., Harlow, UK.
Atectura Breezhaler 125/260	385 µg Indacaterol acetate (125 µg) Mometasone furoate (260 µg)	Lactose monohydrate	Novartis Pharmaceuticals UK Ltd., London, UK.
Pulmicort Turbohaler 400	400 µg Budesonide	None	AstraZeneca UK Ltd., London, UK.
Easyhaler Budesonide 400	400 µg Budesonide	Lactose monohydrate	Orion Pharma UK Ltd., Reading, UK.
Asmanex Twisthaler 400	400 µg Mometasone furoate	Lactose anhydrous	Organon Pharma UK Ltd., London, UK.
Bricanyl Turbohaler 0.5mg	400 µg Terbutaline sulfate	Lactose monohydrate	AstraZeneca UK Ltd., London, UK.
Duaklir Genuair 340/12	407.8 µg Aclidinium bromide (396 µg) Formoterol fumarate dihydrate (11.8 µg)	Lactose monohydrate	Zentiva, Prague, Czech Republic.
Fixkoh Airmaster 50/500	475 µg Fluticasone propionate (432 µg) Salmeterol xinafoate (43 µg)	Lactose monohydrate	Genus Pharmaceuticals, Huddersfield, UK.
Flixotide Accuhaler 500	500 µg Fluticasone propionate	Lactose monohydrate	GlaxoSmithKline UK Ltd., London, UK.
Seretide 500 Accuhaler	507 µg Fluticasone propionate (460 µg) Salmeterol xinafoate (47 µg)	Lactose monohydrate	GlaxoSmithKline UK Ltd., London, UK.
Sereflo Ciphaler 50/500	507 µg Fluticasone propionate (460 µg) Salmeterol xinafoate (47 µg)	Lactose monohydrate	Cipla EU Ltd., Addlestone, UK.
Stalpex 500/50	507 µg Fluticasone propionate (460 µg) Salmeterol xinafoate (47 µg)	Lactose monohydrate	Glenmark Pharmaceuticals Ltd., Mumbai, India.
AirFluSal Forspiro 50/500	510 µg Fluticasone propionate (465 µg) Salmeterol xinafoate (45 µg)	Lactose monohydrate	Sandoz Ltd., Basel, Switzerland.
Fusacomb Easyhaler 50/500	544 µg Fluticasone propionate (496 µg) Salmeterol xinafoate (48 µg)	Lactose monohydrate	Orion Pharma UK Ltd., Reading, UK.
Afrezza	4, 8, or 12 units—Insulin	Fumaryl diketopiperazine (FDKP)	MannKind Corporation, Westlake Village, CA, USA.

^1^ Active Pharmaceutical Ingredient.

## Data Availability

Data sharing is not applicable.

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
