# Peer review of "Inhalation Dosage Forms: A Focus on Dry Powder Inhalers and Their Advancements"

_pharmaceuticals, 2023, doi:10.3390/ph16121658_

Round 1

Reviewer 1 Report

Comments and Suggestions for Authors

The manuscript is very well written and dry powder inhalers are important to review. However, coming up with a very broad topic like this makes it very difficult, if not impossible to write a deep scientific review, although the authors have done a tremendous effort. I recommend the authors to re-write this review by dividing it into several topics. For example, one review can address the influence of excipients on regional deposition using the micronization method. Another review can focus on nanocarriers/microparticles incorporated in spray-dried powder; a third review can focus on various manufacturing methods of DPIs using phospholipid formulations, etc.

Reviewer 2 Report

Comments and Suggestions for Authors

Article is well drafted. But need to consider below few points before processing further. 

Figures - Fig. 2, 3, 4 and 6 - All figs. are with permission or without permission? OR self created? Need details on same. 

Fig. 7, 8 and 9 - Same comments as above.

Fig. 10 - Not needed. Not going to add any specific value to article. 

Table 4 - Need to re-arrangement of data. Try to use below sequence with 4 column table (Inhaler/Medicine name, API name with dose Qty., Carrier and Company name).

Section 3 - Various methods such as electrospinning, TFF and etc. are missing.  Need some commercial examples such as CrystecPharma for SCF and etc.  

Section 6 - Note on digital inhaler device, mobile Apps and etc. is needed.

Need some discussion on how, why and what - new vs old powder processing methods and inhaler devices are useful. 

Add some pictures of currently available marketed DPI devices.    

Comments on the Quality of English Language

NA

Reviewer 3 Report

Comments and Suggestions for Authors

This review extensively analyzes the characteristics, formulation, stability, and manufacturing of dry powder inhalersDPIs, focusing on the advantages of pulmonary administration and the significance of particle size in drug deposition. In addition, the foundation of design and the unique attributes of key DPIs were explored, emphasizing the importance of their optimization, providing valuable insights into the performance of these formulations.

However, there are two points that need to be added

 1. Please provide the description about the biocompatibility of dry powder     inhalersDPIs.

 2. Please supplement relevant literature from the past three years.

Comments on the Quality of English Language

The English expression in this paper is fluent and clear, and the English expression is excellent.

Round 2

Reviewer 1 Report

Comments and Suggestions for Authors

The article is suitable for publication